# A Consensus Anchor-guided Hypergraph Framework for Incomplete Multi-view Clustering

**Yipin Hu** [1]  **Yanxi Liu** [1]  **Fangxi Liu** [1]  **Yanwei Yu** [2]  **Lei Meng** [3]  **Jie Wen** [4]  **Guoqing Chao** [1]

## Abstract

Handling large-scale incomplete multi-view data poses a significant challenge in unsupervised representation learning. While anchor-based strategies have alleviated computational burdens, they typically rely on shallow bipartite graphs restricted to pairwise relations, failing to capture complex high-order correlations among samples. Furthermore, existing methods often treat observed and missing instances indiscriminately, ignoring the distributional shifts that lead to systematic bias in consensus anchor learning. To address these limitations, we propose a novel framework tailored for scalability and robustness, termed **H**ypergraph-**A**ugmented **I**ncomplete **M**ulti-**V**iew **C**lustering (HA-IMVC). Unlike traditional approaches, HA-IMVC constructs a consensus anchor-guided hypergraph that explicitly models group-wise interactions, thereby preserving structural integrity even under high missing rates. Crucially, we incorporate a dual-adaptive reweighting mechanism that calibrates importance at both the view and sample levels. This strategy adaptively penalizes severely incomplete samples to mitigate bias while harmonizing inconsistent views. Extensive experiments on various benchmarks demonstrate that HA-IMVC achieves superior clustering performance and maintains high efficiency, even in scenarios characterized by severe data incompleteness.

---

[1]School of Computer Science and Technology, Harbin Institute of Technology, Weihai, China [2]Faculty of Information Science and Engineering, Ocean University of China, Qingdao, China [3]School of Software, Shandong University, Jinan, China [4]School of Computer Science and Technology, Harbin Institute of Technology, Shenzhen, China. Correspondence to: Guoqing Chao <guoqingchao@hit.edu.cn>.

*Proceedings of the 43rd International Conference on Machine Learning*, Seoul, South Korea. PMLR 306, 2026. Copyright 2026 by the author(s).

## 1. Introduction

Incomplete Multi-View Clustering (IMVC) seeks to group partially observed data into semantically coherent clusters by leveraging complementary information from multiple heterogeneous views (Lin et al., 2021; Wen et al., 2023; Xu et al., 2024; Chao et al., 2025; Yu et al., 2025c;d). In practical applications, the intrinsic complementarity among different views often leads to substantial performance gains. Nevertheless, several critical challenges remain inadequately addressed in real-world IMVC scenarios (Sun et al., 2024; Wang et al., 2011; 2019; Long et al., 2024; Liang et al., 2023; 2024a;b; Li et al., 2025).

In incomplete multi-view data, some samples lack features in certain views, which violates the conventional assumption that each sample is fully observed across all views. To address this issue, a variety of methods have been proposed. Wen et al. (Wen et al., 2024) introduced a diffusion-based framework for missing-view generation, which is further combined with data augmentation strategies to improve clustering performance under high missing rates. Chao et al. (Chao et al., 2024) developed a contrastive learning framework that jointly optimizes missing-view handling, representation learning, and clustering assignment through graph consistency transfer, instance-level attention, and high-confidence guidance. Yu et al. (Yu et al., 2025b) proposed a simple yet effective approach that performs similarity-level imputation and introduces hybrid prototype groups for each view, thereby enhancing multi-scale similarity modeling and clustering performance within a unified framework.

Although completion-based methods can improve performance, explicitly generating missing views on large-scale datasets often incurs prohibitive computational costs. To handle large-scale data while maintaining efficiency, anchor-based approaches offer a promising direction. By simplifying graph construction through representative anchors, these approaches reduce memory consumption and computational overhead. Zhang et al. (Zhang et al., 2024) proposed a cluster structure regularization method to optimize the anchor and cluster assignments jointly, achieving a favorable trade-off between efficiency and accuracy. Liu et al. (Liu et al., 2024a) reviewed anchor generation workflows and proposed

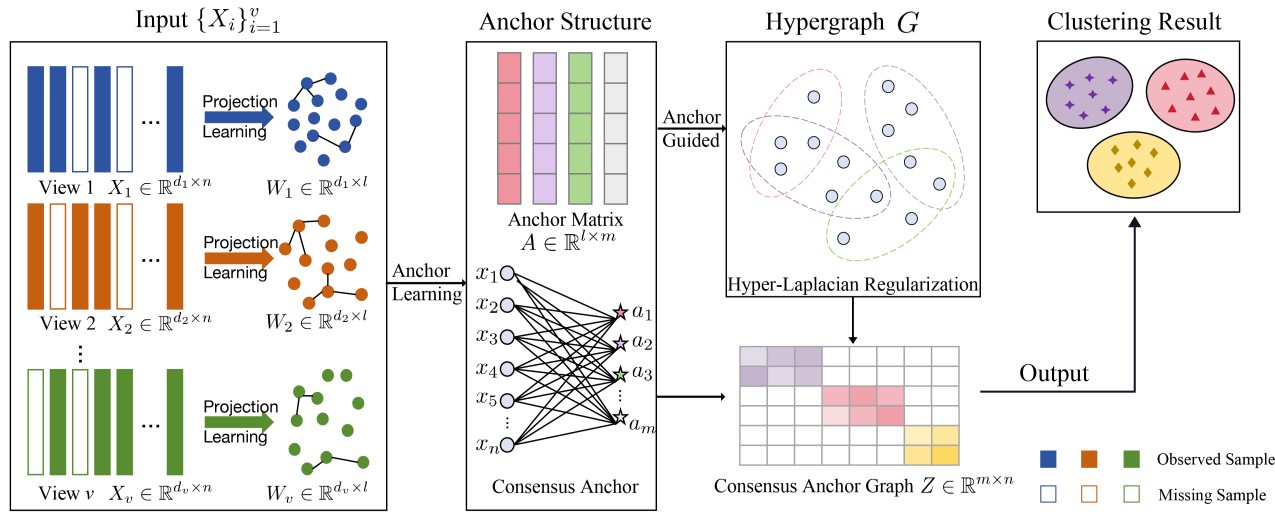

*Figure 1.* Overview of the proposed HA-IMVC framework for incomplete multi-view clustering. The framework first projects incomplete multi-view data $\{X_i\}_{i=1}^v$ into a shared low-dimensional subspace via projection matrices $\{W_i\}_{i=1}^v$, mitigating the impact of missing views. Subsequently, it performs consensus anchor learning to jointly optimize the anchor matrix $A \in \mathbb{R}^{l \times m}$ and the anchor graph $Z \in \mathbb{R}^{m \times n}$. To capture complex high-order correlations, a consensus anchor-guided hypergraph $G$ is constructed, where each anchor induces a hyperedge connecting semantically related samples. Finally, a hypergraph Laplacian regularization term is imposed to enforce group-wise structural consistency, leading to robust and accurate clustering results even under severe data incompleteness.

plug-and-play enhancement strategies to exploit cross-view correlations. Furthermore, Zhang et al. (Zhang et al., 2025) demonstrated that anchor-aware representation learning can capture latent relationships among anchors while preserving scalability in large-scale scenarios.

Although progress has been made in both directions, existing methods still face significant challenges when faced with the complexity of real-world scenarios, mainly in terms of data adaptability and structural robustness. First, at the data level, many methods are built on idealized assumptions that the missingness of the view is random or balanced (Xu et al., 2024; Han et al., 2024). In practice, however, missing samples often exhibit distributional shifts, which introduce systematic biases and consequently impair anchor learning. Moreover, approaches that attempt to address missing views via view recovery are often vulnerable to imputation noise, which can further propagate through the model and degrade clustering performance. Second, at the structural level, most existing anchor-based methods rely on bipartite graphs, which exhibit inherent fragility in incomplete data settings. Since bipartite graphs model only pairwise relations, once a sample is missing in a specific view, its connection to the corresponding anchor is completely severed or becomes unreliable (Li et al., 2022; Mei et al., 2024; Chen et al., 2025). Lacking the structural redundancy provided by high-order correlations, these methods struggle to preserve structural integrity under view-missing scenarios.

To address these limitations, we propose a scalable framework termed Hypergraph-Augmented Incomplete Multi-View Clustering (HA-IMVC). We incorporate a dual-adaptive reweighting mechanism that dynamically calibrates importance at both view and sample levels, effectively mitigating bias from distributional shifts by penalizing severely incomplete instances. Furthermore, we construct a consensus anchor-guided hypergraph to replace fragile bipartite structures. By explicitly modeling group-wise interactions, this design preserves structural integrity even under high missing rates. These components are integrated into a unified model that maintains the efficiency of anchor-based methods. The main contributions are summarized as follows:

1. We propose a **consensus anchor-guided hypergraph framework** that breaks through traditional bipartite graph limitations. By modeling **group-wise interactions**, our method ensures structural robustness against high missing rates.

2. We develop a **dual-adaptive reweighting mechanism** that operates at both the view and sample levels. This strategy mitigates inter-view imbalance and prevents **systematic bias** during anchor learning by penalizing unreliable samples.

3. We systematically validated the effectiveness of the proposed framework through extensive experiments in diverse

datasets. The results demonstrate that HA-IMVC consistently achieves superior performance compared to state-of-the-art methods while maintaining high **computational efficiency** suitable for large-scale applications.

## 2. Related Work

In multi-view clustering, directly constructing similarity graphs in the original high-dimensional feature space is computationally prohibitive and difficult to scale ((He et al., 2025); (Liu et al., 2024b); (Wang et al., 2022b); (Yu et al., 2025a)). To tackle this, the basic anchor graph model introduces view-specific projection matrices to map original data into a shared low-dimensional latent space, while employing a small set of representative anchors to approximate the entire sample set ((Chen et al., 2024); (Sun et al., 2021); (Wang et al., 2022a); (Qin et al., 2025a); (Qin et al., 2025b)). This joint modeling of projection and anchors effectively reduces computational complexity. However, most existing approaches rely on shallow bipartite graphs that restrict interactions to pairwise sample-anchor relations, thereby overlooking complex high-order correlations among samples. Furthermore, in the context of incomplete data, conventional anchor-based methods often struggle to distinguish the distinct characteristics of observed and missing samples, and thus lack explicit filtering or weighting mechanisms to mitigate the impact of low-quality data during consensus learning.

Formally, given $v$ views $\{\mathbf{X}_p\}_{p=1}^v$ with $\mathbf{X}_p \in \mathbb{R}^{d_p \times n}$, each view is projected into a shared $l$-dimensional consensus space via an orthogonal projection matrix $\mathbf{W}_p \in \mathbb{R}^{d_p \times l}$. Let $\mathbf{A} \in \mathbb{R}^{l \times m}$ denote the consensus anchor matrix, $\mathbf{Z} \in \mathbb{R}^{m \times n}$ the anchor graph, and $\beta_p \geq 0$ the view weights with $\sum_{p=1}^v \beta_p = 1$. The basic model can be formulated as:

$$\min_{\{\mathbf{W}_p\}, \mathbf{A}, \mathbf{Z}, \{\beta_p\}} \sum_{p=1}^v \beta_p^2 \left\| \mathbf{X}_p - \mathbf{W}_p \mathbf{A} \mathbf{Z} \right\|_F^2 + \lambda \|\mathbf{Z}\|_F^2,$$

$$\text{s.t.} \quad \mathbf{W}_p^\top \mathbf{W}_p = \mathbf{I}, \mathbf{A}^\top \mathbf{A} = \mathbf{I}, \mathbf{Z} \geq 0, \quad (1)$$

$$\mathbf{Z}^\top \mathbf{1} = \mathbf{1}, \sum_{p=1}^v \beta_p = 1.$$

Final clustering results are then obtained by applying spectral analysis or standard clustering algorithms (e.g., K-Means) on the consensus bipartite graph $\mathbf{Z}$.

Following this paradigm, Wang et al. (Wang et al., 2022a) introduced the anchor graph framework into incomplete multi-view clustering, where unified anchor learning and incomplete anchor graph construction are jointly performed to form a consensus anchor graph. This design preserves cross-view structural consistency while significantly reducing the computational complexity in large-scale IMVC settings. Building upon this idea, Liu et al. (Liu et al., 2022)

unified anchor learning and graph construction within a single framework, and further imposed connectivity constraints to directly generate graphs with explicit cluster structures, enabling one-step clustering without additional post-processing or hyperparameter tuning. Chen et al. (Chen et al., 2024) extended the classical anchor graph framework by introducing an indication matrix to naturally accommodate both complete and incomplete data, and by stacking anchor graphs from multiple views into a tensor with low-rank constraints to explicitly capture high-order cross-view correlations. Ou et al. (Ou et al., 2024) proposed a hierarchical feature descent strategy within the anchor-based model, mapping multi-view features of varying dimensionalities into a unified subspace, and subsequently learning a shared anchor matrix and a consensus bipartite graph to alleviate view discrepancy and enhance scalability. Recently, Qin et al. (Qin et al., 2025a) integrated graph construction, anchor learning, and graph partitioning into a unified framework in which these components mutually reinforce each other. By learning a shared anchor graph to ensure cross-view consistency and explicitly coupling it with symmetric nonnegative matrix factorization, clustering results can be obtained directly.

## 3. Methodology

In this section, we present the Hypergraph-Augmented Incomplete Multi-View Clustering (HA-IMVC) framework, designed to address the structural fragility and systematic bias inherent in existing anchor-based methods. Unlike conventional approaches that rely on shallow bipartite structures, HA-IMVC introduces a consensus anchor-guided hypergraph to transcend the limitations of pairwise modeling. This innovation allows for the explicit capture of group-wise interactions, ensuring structural integrity even under severe data incompleteness. Furthermore, to combat the distributional shifts caused by non-random missingness, we propose a dual-adaptive reweighting mechanism. By dynamically calibrating importance at both view and sample levels, this mechanism effectively suppresses noise and prevents bias propagation. Finally, these components are synergized into a unified objective function to achieve robust and scalable clustering.

### 3.1. Consensus Anchor-guided Hypergraph

To overcome the fragility of pairwise connections in standard bipartite graphs, we construct a consensus anchor-guided hypergraph $\mathcal{H} = (\mathcal{V}, \mathcal{E})$ capturing high-order correlations. The vertex set $\mathcal{V} = \mathcal{V}_s \cup \mathcal{V}_a$ includes $n$ samples and $m$ anchors. Each anchor induces a unique hyperedge $e_j \in \mathcal{E}$ grouping highly correlated samples.

Given the anchor affinity matrix $\mathbf{Z} \in \mathbb{R}^{m \times n}$, we define the weighted incidence matrix $\mathbf{H} \in \mathbb{R}^{(n+m) \times m}$ using the

$T$-nearest anchors $\mathcal{N}_T(i)$ for each sample $i$:

$$
\mathbf{H}_{i,j} = \begin{cases} z_{j,i}, & j \in \mathcal{N}_T(i), \\ 0, & \text{otherwise.} \end{cases} \tag{2}
$$

To explicitly model anchor-sample grouping, we enforce self-loops for anchors by setting $\mathbf{H}_{n+j, j} = 1$ (0 elsewhere). This ensures each hyperedge encapsulates a semantic group of an anchor and its relevant samples.

With vertex and edge degree matrices $\mathbf{D}_v$ and $\mathbf{D}_e$, the normalized hypergraph Laplacian is computed as:

$$
\mathbf{L}_H = \mathbf{I} - \mathbf{D}_v^{-1/2}\,\mathbf{H}\,\mathbf{D}_e^{-1}\,\mathbf{H}^\top\,\mathbf{D}_v^{-1/2}. \tag{3}
$$

We define the augmented representation $\mathbf{Z}_{\text{aug}} = [\mathbf{AZ}, \mathbf{A}]$ and minimize the regularization term $\mathrm{Tr}(\mathbf{Z}_{\text{aug}}\mathbf{L}_H\mathbf{Z}_{\text{aug}}^\top)$. This enforces samples and anchors within the same hyperedge to share similar latent representations, preserving local geometric structures.

## 3.2. Dual-Adaptive Reweighting Mechanism

To mitigate distributional shifts from non-random missingness, we propose a dual-adaptive mechanism operating at both view and sample levels.

**View-level Weighting:** Using the binary mask $\mathbf{S}_p$ for view $p$, we quantify view reliability $\alpha_p$ based on the completeness ratio:

$$
\alpha_p = \frac{\sum_{i=1}^n s_p^{(i)}}{\sum_{u=1}^v \sum_{i=1}^n s_u^{(i)}}, \qquad \text{s.t.} \sum_{p=1}^v \alpha_p = 1. \tag{4}
$$

**Sample-level Weighting:** We evaluate sample quality via a missingness score $q_i \in [0, 1]$, aggregating the weighted missing status:

$$
q_i = 1 - \sum_{p=1}^v \alpha_p\, s_p^{(i)}. \tag{5}
$$

Higher $q_i$ implies lower reliability. We suppress systematic bias via an adaptive weight matrix $\mathbf{M} = \mathrm{Diag}(e^{-\gamma q_i})$ and introduce a missing-rate regularizer $\mathbf{Q} = \mathrm{Diag}(q_i)$ to penalize anchor assignments to unreliable samples.

## 3.3. Unified Objective Function

Integrating these components, the overall HA-IMVC objective is formulated as:

$$
\min_{\beta, \{\mathbf{W}_p\}, \mathbf{A}, \mathbf{Z}} \sum_{p=1}^v \beta_p^2 \left\| \left(\mathbf{W}_p^\top \mathbf{X}_p - \mathbf{AZ}\right) \mathbf{S}_p\, \mathbf{M}^{1/2} \right\|_F^2
$$
$$
+ \lambda_1 \,\mathrm{Tr}(\mathbf{Z}_{\text{aug}}\mathbf{L}_H\mathbf{Z}_{\text{aug}}^\top) + \lambda_2 \,\|\mathbf{ZQ}^{1/2}\|_F^2 \tag{6}
$$

$$
\text{s.t.} \quad \beta_p \geq 0, \sum_{p=1}^v \beta_p = 1; \quad \mathbf{W}_p^\top \mathbf{W}_p = \mathbf{I};
$$
$$
\mathbf{A}^\top \mathbf{A} = \mathbf{I}; \quad \mathbf{Z} \geq 0, \mathbf{Z}^\top \mathbf{1} = \mathbf{1}.
$$

The first term performs *bias-aware latent reconstruction* in the projected subspace, masked by $\mathbf{S}_p$ and weighted by $\mathbf{M}$. The second and third terms enforce *hypergraph consistency* and *reliability-aware sparsity*, respectively, ensuring structural integrity and preventing error propagation from unreliable data.

## 3.4. Optimization

We design an efficient alternating algorithm to optimize the objective function in Eq.(6). In each step, we update one variable while fixing the others.

**Updating $\mathbf{W}_p$:** Fixing other variables, the optimization for the $p$-th view projection matrix $\mathbf{W}_p$ reduces to the following maximization problem:

$$
\max_{\mathbf{W}_p^\top \mathbf{W}_p = \mathbf{I}} \mathrm{Tr}(\mathbf{W}_p^\top \mathbf{G}_p), \quad \mathbf{G}_p = \mathbf{X}_p\,\mathbf{S}_p\mathbf{M}\mathbf{Z}^\top\mathbf{A}^\top. \tag{7}
$$

This is a classic Orthogonal Procrustes problem. Let the SVD of $\mathbf{G}_p$ be $\mathbf{G}_p = \mathbf{U}_p \mathbf{\Sigma}_p \mathbf{V}_p^\top$. The optimal solution is given by:

$$
\mathbf{W}_p^\star = \mathbf{U}_p\mathbf{V}_p^\top. \tag{8}
$$

**Updating $\mathbf{A}$:** Fixing $\mathbf{W}_p$ and $\mathbf{Z}$, and removing constant terms, the optimization problem for $\mathbf{A}$ becomes:

$$
\max_{\mathbf{A}^\top \mathbf{A} = \mathbf{I}} \mathrm{Tr}(\mathbf{A}^\top \mathbf{P}), \quad \mathbf{P} = \sum_{p=1}^v \beta_p^2\,\mathbf{W}_p^\top\mathbf{X}_p\,\mathbf{S}_p\mathbf{M}\mathbf{Z}^\top. \tag{9}
$$

Similar to the $\mathbf{W}_p$ update, let the SVD of $\mathbf{P}$ be $\mathbf{P} = \mathbf{U}_A\mathbf{\Sigma}_A\mathbf{V}_A^\top$. The closed-form update is $\mathbf{A}^\star = \mathbf{U}_A\mathbf{V}_A^\top$.

**Updating $\mathbf{Z}$:** We partition the hypergraph Laplacian as $\mathbf{L}_H = \begin{bmatrix} \mathbf{L}_{dd} & \mathbf{L}_{da} \\ \mathbf{L}_{ad} & \mathbf{L}_{aa} \end{bmatrix}$, where $\mathbf{L}_{ad} = \mathbf{L}_{da}^\top$. Given the definition $\mathbf{Z}_{\text{aug}} = [\mathbf{AZ}, \mathbf{A}]$ and the constraint $\mathbf{A}^\top \mathbf{A} = \mathbf{I}$, the regularization term simplifies as:

$$
\mathrm{Tr}(\mathbf{Z}_{\text{aug}}\mathbf{L}_H\mathbf{Z}_{\text{aug}}^\top) = \mathrm{Tr}(\mathbf{Z}\mathbf{L}_{dd}\mathbf{Z}^\top) + 2\,\mathrm{Tr}(\mathbf{Z}^\top\mathbf{L}_{ad}) + \text{const.} \tag{10}
$$

**Algorithm 1** $\mathbf{Z}$-update via FISTA

**Require:** $\mathbf{K} \succeq 0$, $\mathbf{J}$; initial $\mathbf{Z}^{(0)} = \mathbf{Z}^{(1)}$, $t_1 = 1$; Lipschitz constant $\mathcal{L} \geq 2\lambda_{\max}(\mathbf{K})$; tolerance $\varepsilon$.
1: **while** not converged **do**
2:    $t_{k+1} \leftarrow \left(1 + \sqrt{1 + 4t_k^2}\right)/2$;
3:    $\mathbf{Y}^{(k)} \leftarrow \mathbf{Z}^{(k)} + \frac{t_k - 1}{t_{k+1}}\left(\mathbf{Z}^{(k)} - \mathbf{Z}^{(k-1)}\right)$;
4:    $\widetilde{\mathbf{Z}} \leftarrow \mathbf{Y}^{(k)} - \frac{1}{\mathcal{L}}\left(2\mathbf{Y}^{(k)}\mathbf{K} - 2\mathbf{J}\right)$;
5:    **for** each column $j = 1, \ldots, n$ **do**
6:       $\mathbf{Z}_{:,j}^{(k+1)} \leftarrow \Pi_\Delta(\widetilde{\mathbf{Z}}_{:,j})$;
7:    **end for**
8:    **if** $\frac{\|\mathbf{Z}^{(k+1)} - \mathbf{Z}^{(k)}\|_F}{\max\{1, \|\mathbf{Z}^{(k)}\|_F\}} < \varepsilon$ **then break**;
9: **end while**
**Ensure:** Optimized $\mathbf{Z}$.

---

**Algorithm 2** Alternating Optimization for HA-IMVC

**Require:** Multi-view data $\{\mathbf{X}_p\}_{p=1}^v$, masks $\{\mathbf{S}_p\}_{p=1}^v$; Parameters $\lambda_1, \lambda_2, T, m, k$.
1: Initialize $\mathbf{M}$, $\mathbf{Q}$ based on missing rates.
2: Initialize $\mathbf{Z}$ and $\{\mathbf{W}_p, \mathbf{A}\}$ randomly or via K-Means.
3: **repeat**
4:    Update projections $\{\mathbf{W}_p\}_{p=1}^v$ via Eq.(7);
5:    Update anchor matrix $\mathbf{A}$ via Eq.(9);
6:    Update anchor graph $\mathbf{Z}$ via Algorithm 1;
7:    Update view weights $\{\beta_p\}_{p=1}^v$ via Eq.(13);
8:    Update hypergraph structure $\mathbf{L}_H$ via Eq.(3);
9: **until** convergence;
**Ensure:** Clustering result by applying K-Means on $\mathbf{Z}$.

Consequently, the $\mathbf{Z}$-subproblem can be reformulated as a Quadratic Programming (QP) problem:

$$\min_{\mathbf{Z} \in \mathbb{R}^{m \times n}} \quad f(\mathbf{Z}) = \mathrm{Tr}(\mathbf{Z}\mathbf{K}\mathbf{Z}^\top) - 2\,\mathrm{Tr}(\mathbf{Z}^\top \mathbf{J})$$
$$\text{s.t.} \quad \mathbf{Z} \geq \mathbf{0}, \qquad \mathbf{Z}^\top \mathbf{1} = \mathbf{1}, \tag{11}$$

where the auxiliary matrices are defined as:

$$\mathbf{K} = \sum_{p=1}^v \beta_p^2 \mathbf{S}_p \mathbf{M} + \lambda_1 \mathbf{L}_{dd} + \lambda_2 \mathbf{Q},$$
$$\mathbf{J} = \sum_{p=1}^v \beta_p^2 \mathbf{A}^\top \mathbf{W}_p^\top \mathbf{X}_p \mathbf{S}_p \mathbf{M} - \lambda_1 \mathbf{L}_{ad}. \tag{12}$$

We solve Eq.(11) using FISTA (Beck & Teboulle, 2009) with gradient $\nabla f(\mathbf{Z}) = 2\mathbf{Z}\mathbf{K} - 2\mathbf{J}$. The stepsize is set as $1/\mathcal{L}$ where $\mathcal{L} \geq 2\lambda_{\max}(\mathbf{K})$. Each gradient step is followed by a column-wise projection onto the probability simplex $\Delta = \{z \in \mathbb{R}^m : z \geq 0, \mathbf{1}^\top z = 1\}$. The procedure is summarized in Algorithm 1.

**Updating $\boldsymbol{\beta}$:** The optimization w.r.t. the view weights $\boldsymbol{\beta}$ is:

$$\min_{\beta \geq 0, \sum \beta_p = 1} \sum_{p=1}^v \beta_p^2 R_p^2, \tag{13}$$

where $R_p = \left\|\left(\mathbf{W}_p^\top \mathbf{X}_p - \mathbf{A}\mathbf{Z}\right)\mathbf{S}_p \mathbf{M}^{1/2}\right\|_F$ represents the reconstruction error of the $p$-th view. The closed-form solution is:

$$\beta_p^\star = \frac{1/R_p^2}{\sum_{u=1}^v (1/R_u^2)}. \tag{14}$$

**Updating $\mathbf{L}_H$:** We dynamically rebuild the anchor-guided hypergraph based on the updated $\mathbf{Z}$. For each sample $i$, we identify its $T$-nearest anchors based on the values in $\mathbf{Z}_{:,i}$, update the incidence matrix $\mathbf{H}$, and recompute $\mathbf{L}_H$ via Eq.(3).

### 3.5. Complexity Analysis

The time complexity of HA-IMVC is primarily dictated by the iterative updates of the anchor graph and the dynamic construction of the hypergraph. Specifically, the time complexity per iteration is dominated by the gradient computations in FISTA and the sorting operations required to identify $T$-nearest anchors, totaling $\mathcal{O}(n(m \log m + ml))$. In contrast, the SVD-based updates for projection matrices and anchors depend only on feature dimensions and latent sizes, rendering them computationally negligible given $l, m \ll n$.

Regarding space complexity, the dominant terms are the storage of the consensus anchor graph $\mathbf{Z}$ and the sparse hypergraph incidence matrix $\mathbf{H}$, which require $\mathcal{O}(n(m + T))$ space. Notably, the storage for model parameters (e.g., $\mathbf{W}_p, \mathbf{A}$) is independent of $n$. Since $m$, $l$, and $T$ are small constants relative to the sample size $n$, both time and space complexities scale linearly, i.e., $\mathcal{O}(n)$, ensuring the framework's scalability for large-scale incomplete multi-view datasets.

## 4. Experiments

### 4.1. Datasets and Baselines

We conduct experiments on six widely-used multi-view datasets. The detailed statistics of these datasets are summarized in Table 1.

*Table 1.* Statistics of benchmark datasets

| Dataset | Samples | Views | Classes |
|---------|---------|-------|---------|
| NGs | 500 | 3 | 5 |
| Caltech101-20 | 2,396 | 6 | 20 |
| BDGP | 2,500 | 3 | 5 |
| CCV | 6,773 | 3 | 20 |
| Animal | 11,673 | 4 | 50 |
| MNIST | 60,000 | 3 | 10 |

HA-IMVC is compared with the following state-of-the-art multi-view clustering approaches, categorized into two groups:

- **Non-anchor-based methods:** BSV (Ng et al., 2001), HCP-IMSC (Li et al., 2022), UOMVSC (Tang et al., 2023), and SCSL (Liu et al., 2024c).

- **Anchor-based methods:** EMKMC (Yang et al., 2023), FastMICE (Huang et al., 2023), FDAGF (Zhang et al., 2023), and MVSC-HFD (Ou et al., 2024).

### 4.2. Results and Discussions

We employ three widely adopted metrics to evaluate clustering results: Accuracy (ACC), Normalized Mutual Information (NMI), and Purity (PUR). To mitigate randomness, each experiment is repeated 20 times, reporting the mean and standard deviation. Tables 2 and 3 present the clustering performance of all methods under missing rates of 30%, 50%, and 70%. Methods that are unable to run due to out-of-memory errors are denoted as N/A. Based on the experimental results, we draw the following conclusions:

- **Superior Performance:** HA-IMVC outperforms most baseline algorithms in various datasets and missing rates. In particular, it consistently achieves the best performance in the NGs and BDGP datasets. In the more challenging Caltech101-20 and Animal datasets, HA-IMVC remains highly competitive, ranking first or second in most metrics. Crucially, even with a missing rate as high as 70%, HA-IMVC maintains robust performance, demonstrating its effectiveness in handling severe data incompleteness.

- **Scalability:** Conventional non-anchor-based methods, such as HCP-IMSC and SCSL, fail to scale to larger datasets like MNIST due to high computational complexity. In contrast, HA-IMVC works reliably in large-scale scenarios while achieving superior accuracy (e.g., 98.70% ACC in MNIST at 30% missing rate), highlighting its practicality for real-world applications.

### 4.3. Time Comparison

Figure 2 reports the comparison of the running time between HA-IMVC and baseline methods in six datasets. Note that the vertical axis utilizes a logarithmic scale ($\log_2$) to handle the significant variance in running times. As observed, HA-IMVC consistently demonstrates superior efficiency, requiring significantly less time than most competitors. Notably, on large-scale datasets such as Animal and MNIST, several non-anchor-based methods (e.g., HCP-IMSC, SCSL) failed to execute due to memory constraints (indicated by

missing bars), whereas HA-IMVC completed the tasks efficiently. This empirical evidence supports our theoretical complexity analysis, confirming that HA-IMVC achieves a favorable balance between high clustering accuracy and computational scalability.

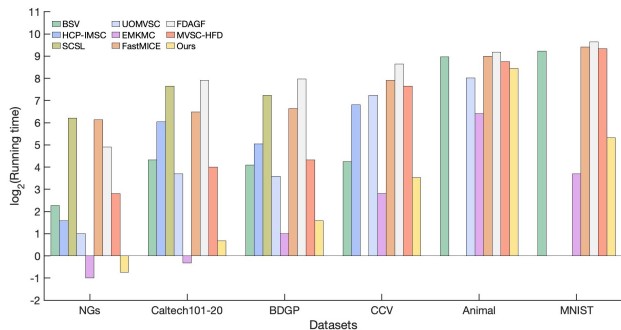

*Figure 2.* Running time comparison of different methods on six datasets.

### 4.4. Ablation Study

To evaluate the contributions of the Hypergraph (HG) module and the Reweighting (RW) module, we compare the proposed framework with the Base (bipartite), Reweigh (RW), HG (hypergraph), and HA-IMVC. Specifically, "Base (bipartite)" relies solely on the conventional pairwise anchor-sample bipartite graph; "Reweigh" denotes the variant where the hypergraph structure is removed, leaving only the reweighting mechanism; "HG (hypergraph)" represents the model with only the hypergraph structure but without the reweighting mechanism; and "HA-IMVC" denotes our complete model, which constructs an anchor-guided hypergraph to model high-order correlations.

As shown in Table 4, the HA-IMVC model consistently outperforms the other three variants across all six datasets and three missingness levels. These results validate that the hypergraph structure effectively captures group-wise interactions among samples sharing common anchors. By enforcing high-order structural consistency and the reweighting mechanism, the framework significantly enhances clustering robustness, particularly in scenarios with severe data incompleteness and cross-view noise.

### 4.5. Convergence

To thoroughly verify and comprehensively evaluate the convergence performance of the proposed algorithm, we meticulously track and record the dynamic variations of the objective function value throughout the entire iterative process. On a theoretical level, owing to the adoption of an alternating optimization strategy, each subproblem is guaranteed to be solved optimally under the current variable constraints during each iteration loop. Consequently, the total objective

*Table 2.* Clustering results of different methods on NGs, Caltech101-20, and BDGP Datasets (Mean ± Std).

| Method | NGs | | | | | | | | |
|---|---|---|---|---|---|---|---|---|---|
| | **30%** | | | **50%** | | | **70%** | | |
| | ACC | NMI | PUR | ACC | NMI | PUR | ACC | NMI | PUR |
| BSV | 39.07±1.58 | 19.43±0.87 | 39.68±1.60 | 33.15±1.42 | 13.80±1.39 | 34.03±1.14 | 25.74±0.83 | 6.71±0.75 | 26.28±0.85 |
| HCP-IMSC | 93.40±0.00 | 80.31±0.00 | 93.40±0.00 | 89.00±0.00 | 71.46±0.00 | 89.00±0.00 | 85.10±0.00 | 60.49±0.00 | 85.10±0.00 |
| SCSL | 60.77±0.73 | 42.52±0.10 | 64.60±0.22 | 38.72±0.65 | 39.92±0.22 | 36.86±0.19 | 29.11±0.16 | 38.31±0.11 | 30.18±0.14 |
| UOMVSC | 73.17±0.01 | 67.02±0.00 | 73.93±0.00 | 73.04±0.01 | 65.81±0.00 | 72.88±0.00 | 71.11±0.00 | 60.84±0.00 | 70.74±0.00 |
| EMKMC | 45.13±0.00 | 38.01±0.00 | 45.15±0.00 | 44.78±0.00 | 37.62±0.00 | 44.76±0.00 | 42.53±0.00 | 34.57±0.00 | 41.30±0.00 |
| FastMICE | 40.42±0.03 | 18.23±0.09 | 41.37±0.08 | 39.23±0.05 | 16.34±0.07 | 40.41±0.07 | 37.23±0.03 | 14.25±0.17 | 38.23±0.07 |
| FDAGF | 53.33±0.00 | 34.75±0.00 | 54.99±0.00 | 52.93±0.00 | 33.82±0.00 | 54.10±0.00 | 52.82±0.00 | 33.21±0.00 | 54.32±0.00 |
| MVSC-HFD | 46.76±6.46 | 24.01±5.77 | 47.88±6.24 | 42.40±2.78 | 17.87±2.62 | 42.92±3.21 | 37.80±2.62 | 12.38±1.27 | 38.12±2.80 |
| **Ours** | **94.20±0.00** | **83.53±0.00** | **94.20±0.00** | **91.20±0.00** | **77.38±0.00** | **91.20±0.00** | **87.42±0.09** | **69.37±0.13** | **87.42±0.09** |

| Method | Caltech101-20 | | | | | | | | |
|---|---|---|---|---|---|---|---|---|---|
| | **30%** | | | **50%** | | | **70%** | | |
| | ACC | NMI | PUR | ACC | NMI | PUR | ACC | NMI | PUR |
| BSV | 39.71±3.14 | 53.04±1.05 | 68.77±1.13 | 36.86±3.84 | 49.04±1.68 | 65.32±1.46 | 33.01±2.75 | 43.96±1.06 | 61.05±1.11 |
| HCP-IMSC | 46.44±2.21 | 50.38±0.99 | 66.76±0.59 | 42.66±1.82 | 50.97±0.96 | 67.07±0.92 | 41.13±1.48 | 50.50±0.66 | 66.42±0.69 |
| SCSL | 43.84±1.66 | 57.30±0.68 | 75.50±0.61 | 43.39±1.83 | 55.83±0.57 | 72.78±0.84 | 45.05±1.64 | 52.81±0.73 | 70.04±0.77 |
| UOMVSC | 44.98±0.01 | 60.17±0.00 | 75.68±0.07 | 41.79±0.00 | 57.48±0.01 | 72.63±0.00 | 38.57±0.00 | 53.79±0.00 | 68.57±0.02 |
| EMKMC | 30.87±0.00 | 32.64±0.00 | 56.12±0.00 | 28.57±0.00 | 31.47±0.00 | 54.01±0.00 | 27.50±0.00 | 31.02±0.00 | 53.45±0.00 |
| FastMICE | 34.27±2.02 | 59.32±0.86 | 75.24±0.57 | 43.12±0.31 | 57.23±0.31 | 73.25±0.52 | 34.50±1.22 | 53.34±0.23 | 70.26±0.84 |
| FDAGF | 41.22±2.46 | 49.23±0.07 | 67.25±2.61 | 43.12±2.56 | 50.15±0.00 | 69.36±1.96 | 40.49±3.72 | 48.13±0.02 | 66.20±2.84 |
| MVSC-HFD | 51.09±3.17 | 45.63±1.94 | 64.69±1.79 | 48.01±2.53 | 42.81±1.58 | 63.10±1.81 | 41.94±3.88 | 38.25±1.61 | 58.18±2.33 |
| **Ours** | **55.23±1.99** | 59.81±0.64 | **76.34±0.44** | **55.34±2.42** | 57.27±0.60 | **73.34±0.42** | **51.53±1.99** | **55.91±0.45** | **73.72±0.42** |

| Method | BDGP | | | | | | | | |
|---|---|---|---|---|---|---|---|---|---|
| | **30%** | | | **50%** | | | **70%** | | |
| | ACC | NMI | PUR | ACC | NMI | PUR | ACC | NMI | PUR |
| BSV | 36.22±0.85 | 21.40±0.92 | 38.02±0.92 | 32.88±0.66 | 16.76±0.71 | 33.82±0.69 | 31.42±0.69 | 14.77±0.75 | 32.69±0.71 |
| HCP-IMSC | 34.28±0.36 | 12.76±0.02 | 36.38±0.01 | 32.44±0.20 | 12.37±0.04 | 35.35±0.01 | 33.25±0.04 | 11.72±0.02 | 34.75±0.02 |
| SCSL | 29.08±0.89 | 9.19±2.69 | 30.28±0.11 | 30.89±1.96 | 6.72±1.94 | 31.29±1.96 | 29.69±1.88 | 4.71±2.54 | 30.09±1.88 |
| UOMVSC | 38.97±0.03 | 15.56±0.00 | 41.69±0.01 | 36.34±0.00 | 14.24±0.01 | 39.36±0.04 | 33.29±0.00 | 13.97±0.00 | 35.16±0.00 |
| EMKMC | 31.46±0.00 | 8.34±0.00 | 32.76±0.00 | 31.05±0.00 | 6.77±0.00 | 31.34±0.00 | 28.53±0.00 | 6.78±0.00 | 29.21±0.00 |
| FastMICE | 35.05±0.00 | 12.78±0.00 | 33.27±0.00 | 34.05±0.00 | 12.18±0.00 | 32.16±0.00 | 33.14±0.00 | 11.66±0.00 | 31.37±0.00 |
| FDAGF | 48.65±3.61 | 25.65±5.05 | 49.18±2.98 | 46.38±2.41 | 25.15±4.64 | 48.71±2.28 | 43.04±3.52 | 22.12±3.17 | 42.68±0.94 |
| MVSC-HFD | 39.06±1.06 | 13.37±0.83 | 39.27±0.87 | 34.89±2.78 | 9.77±1.43 | 35.47±2.91 | 32.89±2.55 | 8.26±0.77 | 33.71±2.17 |
| **Ours** | **50.57±0.05** | **26.43±0.09** | **50.83±0.04** | **48.52±0.05** | **25.22±0.13** | **49.06±0.05** | **46.80±0.01** | **22.36±0.01** | **47.32±0.00** |

function value of the system is mathematically guaranteed to possess a favorable monotonically non-increasing property.

This theoretical expectation is fully substantiated by the empirical results. As illustrated in Figure 3, the objective function value curve exhibits a clear and strictly monotonic decreasing trend. Notably, during the initial few rounds of iteration, the convergence curve undergoes a sharp drop, indicating that the algorithm can swiftly capture the core optimization direction within the solution space, thereby demonstrating exceptional search efficiency. Subsequently, the curve flattens out and rapidly stabilizes, forming a steady horizontal asymptote. Specifically, the algorithm exhibits an outstanding fast-convergence characteristic, typically reaching the convergence threshold within fewer than 20 iterations. Such consistent behavior across diverse experimental scenarios powerfully validates the robustness of our approach. In summary, the HA-IMVC algorithm not only offers rigorous theoretical convergence guarantees but also efficiently converges to a high-quality local optimum with remarkably low computational complexity and time overhead in practical applications.

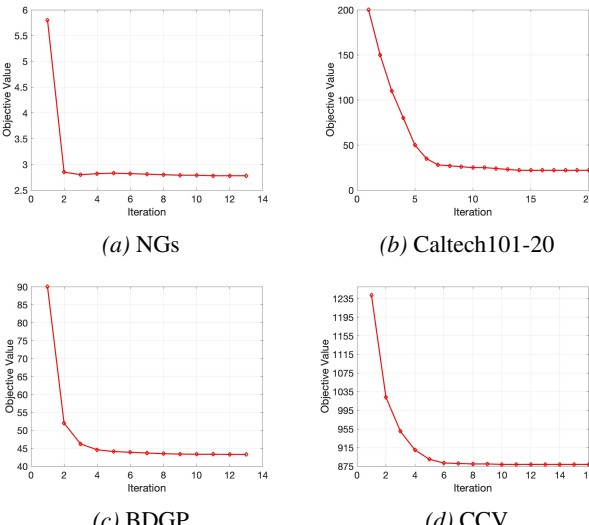

*(a)* NGs
*(b)* Caltech101-20

*(c)* BDGP
*(d)* CCV

*Figure 3.* Convergence curves of the objective function value on different datasets: (a) NGs, (b) Caltech101-20, (c) BDGP, and (d) CCV.

*Table 3.* Clustering results of different methods on CCV, Animal, and MNIST (Mean $\pm$ Std).

| Method | CCV | | | | | | | | |
|---|---|---|---|---|---|---|---|---|---|
| | **30%** | | | **50%** | | | **70%** | | |
| | ACC | NMI | PUR | ACC | NMI | PUR | ACC | NMI | PUR |
| BSV | 19.26±0.06 | 9.86±0.05 | 17.40±0.08 | 18.35±0.05 | 13.90±0.03 | 20.56±0.04 | 17.38±0.04 | 12.88±0.02 | 19.35±0.03 |
| HCP-IMSC | 10.78±0.07 | 10.76±0.11 | 10.67±0.09 | 10.03±0.09 | 9.91±0.15 | 10.15±0.05 | 9.42±0.07 | 9.13±0.07 | 9.95±0.15 |
| SCSL | N/A | N/A | N/A | N/A | N/A | N/A | N/A | N/A | N/A |
| UOMVSC | 10.91±0.01 | 10.89±0.04 | 10.80±0.04 | 10.21±0.00 | 10.86±0.01 | 9.85±0.00 | 9.45±0.01 | 9.79±0.02 | 9.21±0.02 |
| EMKMC | 11.25±0.00 | 5.77±0.00 | 15.46±0.00 | 10.84±0.00 | 5.73±0.00 | 15.23±0.00 | 10.44±0.00 | 2.98±0.00 | 14.35±0.00 |
| FastMICE | 20.12±0.23 | 8.23±0.09 | 21.37±0.08 | 19.23±0.15 | 7.34±0.07 | 20.41±0.07 | 15.23±0.03 | 4.25±0.17 | 18.23±0.07 |
| FDAGF | 10.50±1.03 | 5.97±0.03 | 19.80±3.65 | 10.12±3.41 | 5.56±0.05 | 19.18±3.36 | 9.45±2.15 | 5.16±0.06 | 18.72±2.78 |
| MVSC-HFD | 20.58±0.00 | 14.41±0.00 | 23.40±0.00 | 18.74±0.00 | 12.89±0.00 | 21.79±0.00 | 16.67±0.00 | 11.13±0.04 | 19.43±0.00 |
| **Ours** | **23.25±0.17** | **16.58±0.04** | **25.96±0.05** | **21.63±0.05** | **15.48±0.04** | **25.09±0.05** | **18.25±0.05** | **13.82±0.07** | **22.02±0.08** |

| Method | Animal | | | | | | | | |
|---|---|---|---|---|---|---|---|---|---|
| | **30%** | | | **50%** | | | **70%** | | |
| | ACC | NMI | PUR | ACC | NMI | PUR | ACC | NMI | PUR |
| BSV | 15.32±0.08 | 10.11±0.04 | 16.38±0.12 | 14.69±0.07 | 9.28±0.05 | 15.55±0.03 | 13.58±0.09 | 7.95±0.09 | 14.68±0.03 |
| HCP-IMSC | N/A | N/A | N/A | N/A | N/A | N/A | N/A | N/A | N/A |
| SCSL | N/A | N/A | N/A | N/A | N/A | N/A | N/A | N/A | N/A |
| UOMVSC | 13.45±1.79 | 11.56±3.64 | 17.78±1.36 | 12.13±2.13 | 10.45±2.79 | 16.42±1.63 | 10.67±2.31 | 9.37±1.56 | 14.41±3.14 |
| EMKMC | 11.45±0.00 | 6.88±0.00 | 11.32±0.00 | 10.43±0.00 | 6.82±0.00 | 10.29±0.00 | 11.14±0.00 | 5.85±0.00 | 11.28±0.00 |
| FastMICE | 9.08±0.00 | 8.18±0.00 | 11.23±0.00 | 13.67±0.31 | 8.17±0.00 | 11.34±0.00 | 8.87±0.00 | 7.58±0.00 | 10.93±0.00 |
| FDAGF | 15.71±0.14 | 9.15±0.25 | 16.55±0.47 | 13.67±0.31 | 7.11±0.14 | 15.11±0.25 | 12.52±0.65 | 6.44±0.23 | 14.51±0.26 |
| MVSC-HFD | **17.60±0.22** | **13.19±0.18** | 20.52±0.10 | 16.13±0.25 | **11.85±0.20** | 19.59±0.13 | 16.15±0.45 | 11.09±0.46 | 19.22±0.57 |
| **Ours** | 17.30±0.00 | 12.65±0.18 | **20.56±0.19** | **16.79±0.07** | 11.59±0.02 | **19.85±0.04** | **16.46±0.01** | **11.21±0.01** | **19.38±0.02** |

| Method | MNIST | | | | | | | | |
|---|---|---|---|---|---|---|---|---|---|
| | **30%** | | | **50%** | | | **70%** | | |
| | ACC | NMI | PUR | ACC | NMI | PUR | ACC | NMI | PUR |
| BSV | 75.88±0.49 | 74.67±0.19 | 78.58±0.33 | 66.49±0.69 | 65.39±0.27 | 69.45±0.49 | 60.69±0.52 | 58.93±0.22 | 62.66±0.38 |
| HCP-IMSC | N/A | N/A | N/A | N/A | N/A | N/A | N/A | N/A | N/A |
| SCSL | N/A | N/A | N/A | N/A | N/A | N/A | N/A | N/A | N/A |
| UOMVSC | N/A | N/A | N/A | N/A | N/A | N/A | N/A | N/A | N/A |
| EMKMC | 71.21±0.30 | 70.88±0.22 | 72.32±0.43 | 70.43±0.23 | 70.82±0.43 | 71.29±0.20 | 70.14±0.00 | 70.25±0.25 | 71.18±0.30 |
| FastMICE | 97.45±0.00 | 96.08±0.00 | 97.89±0.00 | 97.53±0.01 | 95.52±0.00 | 97.68±0.00 | 97.24±0.01 | 95.05±0.00 | 96.89±0.01 |
| FDAGF | 98.64±0.15 | 96.12±0.47 | 97.08±0.95 | 98.23±0.21 | 95.89±0.11 | 97.10±0.23 | 98.05±0.36 | 95.01±0.24 | 96.85±0.34 |
| MVSC-HFD | 75.88±4.86 | 74.67±1.92 | 78.58±3.34 | 66.50±6.99 | 65.39±2.77 | 69.45±4.88 | 60.69±5.21 | 58.93±2.19 | 62.66±3.76 |
| **Ours** | **98.70±0.00** | **96.27±0.00** | **98.42±0.00** | **98.59±0.00** | **96.59±0.00** | **98.42±0.00** | **98.36±0.00** | **95.16±0.00** | **98.32±0.00** |

## 4.6. Parameter Sensitivity Analysis

For HA-IMVC, the anchor number $m$ and the projection dimension $l$ are tuned in $\{k, 2k, \ldots, 10k\}$, while the nearest neighbor count $T$ is selected from $\{5, 10, 15, 20\}$.

To assess robustness, we perform a grid search for $\lambda_1$ and $\lambda_2$ within $\{10^{-5}, \ldots, 10^1\}$ on four datasets (50% missing rate). As shown in Figure 4, the model maintains consistently high accuracy across a broad range (e.g., $\lambda_1 \in [10^{-5}, 10^{-2}]$, $\lambda_2 \in [10^{-5}, 10^{-1}]$), with performance dropping only at extremes. This confirms that HA-IMVC is insensitive to hyperparameter variations and stable for practical deployment.

As shown in Table 5, we evaluated the impact of $\gamma$. We assessed the ACC of HA-IMVC under the 50% missing rate. The experimental results indicate that model performance peaks at a 50% missing rate. This strongly validates the effectiveness of this mechanism in handling distribution shifts in incomplete views. When $\gamma$ is close to 0 or exceeds 1.5, performance declines to varying degrees for various reasons. This reflects the critical importance of striking a balance between noise suppression and information preservation.

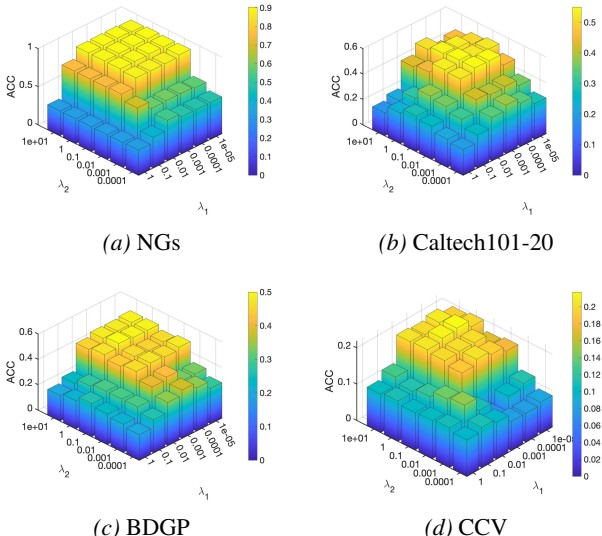

*(a)* NGs

*(b)* Caltech101-20

*(c)* BDGP

*(d)* CCV

*Figure 4.* Clustering accuracy (ACC) with different parameter combinations of $\lambda_1$ and $\lambda_2$ across four datasets: (a) NGs, (b) Caltech101-20, (c) BDGP, and (d) CCV.

*Table 4.* Ablation Study on the Effectiveness of the Hypergraph Module and the Reweighting module.

| Rate | Variant | NGs | | | Caltech101-20 | | | BDGP | | | CCV | | | Animal | | | MNIST | | |
|------|---------|-----|-----|-----|-----|-----|-----|-----|-----|-----|-----|-----|-----|-----|-----|-----|-----|-----|-----|
| | | ACC | NMI | PUR | ACC | NMI | PUR | ACC | NMI | PUR | ACC | NMI | PUR | ACC | NMI | PUR | ACC | NMI | PUR |
| 30% | Base | 88.50 | 74.20 | 88.50 | 48.23 | 50.54 | 68.33 | 46.73 | 23.64 | 46.63 | 19.34 | 15.84 | 22.60 | 13.68 | 10.64 | 16.98 | 97.65 | 94.84 | 97.13 |
| | HG | 92.15 | 80.10 | 92.15 | 50.86 | 51.63 | 69.84 | 48.73 | 24.64 | 48.75 | 20.74 | 16.45 | 24.80 | 15.23 | 11.23 | 17.90 | 98.44 | 95.74 | 97.96 |
| | RW | 90.80 | 77.44 | 90.80 | 51.26 | 53.48 | 71.22 | 48.26 | 24.32 | 48.20 | 21.71 | 16.17 | 24.14 | 15.65 | 11.62 | 18.31 | 98.33 | 95.43 | 97.89 |
| | **HA-IMVC** | **94.20** | **83.53** | **94.20** | **55.23** | **58.51** | **76.34** | **50.57** | **26.43** | **50.83** | **23.25** | **16.58** | **25.96** | **17.30** | **12.65** | **20.56** | **98.60** | **95.87** | **98.32** |
| 50% | Base | 86.30 | 70.15 | 86.30 | 47.64 | 48.57 | 67.44 | 40.68 | 19.43 | 42.28 | 16.74 | 11.63 | 18.35 | 11.64 | 8.90 | 16.36 | 97.45 | 94.11 | 96.90 |
| | HG | 89.95 | 75.40 | 89.95 | 49.64 | 49.37 | 69.06 | 43.62 | 21.53 | 43.63 | 18.02 | 12.97 | 21.53 | 12.53 | 10.85 | 17.34 | 98.32 | 95.23 | 97.81 |
| | RW | 89.40 | 73.33 | 89.40 | 52.01 | 50.38 | 69.21 | 44.31 | 22.68 | 45.31 | 18.35 | 13.82 | 21.90 | 14.83 | 10.80 | 17.61 | 98.21 | 95.24 | 97.89 |
| | **HA-IMVC** | **91.20** | **77.38** | **91.20** | **55.34** | **56.07** | **73.34** | **48.52** | **25.22** | **49.06** | **21.63** | **15.48** | **25.09** | **16.79** | **11.59** | **19.85** | **98.49** | **95.59** | **98.32** |
| 70% | Base | 80.20 | 58.40 | 80.20 | 45.62 | 46.35 | 64.11 | 38.56 | 14.58 | 40.76 | 17.53 | 11.87 | 19.35 | 13.64 | 9.64 | 14.75 | 97.11 | 93.87 | 96.34 |
| | HG | 85.10 | 66.50 | 85.10 | 47.36 | 49.74 | 68.85 | 44.64 | 17.83 | 44.53 | 17.90 | 13.25 | 21.19 | 15.64 | 10.33 | 15.96 | 97.89 | 94.67 | 97.43 |
| | RW | 83.60 | 62.93 | 83.60 | 47.56 | 50.12 | 70.23 | 42.98 | 18.77 | 43.23 | 18.08 | 13.16 | 21.35 | 15.35 | 10.61 | 16.93 | 98.12 | 94.95 | 97.89 |
| | **HA-IMVC** | **87.42** | **69.37** | **87.42** | **51.53** | **53.91** | **73.72** | **46.80** | **21.36** | **47.32** | **18.25** | **13.82** | **22.02** | **16.46** | **11.21** | **19.38** | **98.36** | **95.16** | **98.32** |

*Table 5.* Sensitivity analysis of $\gamma$ in terms of ACC (%) under 50% missing rate.

| $\gamma$ Value | NGs | Caltech101-20 | BDGP |
|----------------|-----|---------------|------|
| 0.2 | 79.14 | 38.53 | 33.65 |
| 0.5 | 84.12 | 42.54 | 39.75 |
| **0.8 (Optimal)** | **91.20** | **55.34** | **48.52** |
| 1.1 | 85.23 | 46.37 | 41.64 |
| 1.4 | 81.34 | 41.74 | 37.37 |
| 1.7 | 76.45 | 36.75 | 31.53 |

## 5. Conclusion

In this paper, we proposed HA-IMVC, a scalable framework tailored for robust unsupervised representation learning. To overcome the structural fragility of conventional bipartite graphs, HA-IMVC constructs a consensus anchor-guided hypergraph that explicitly models group-wise interactions between anchors and samples. Furthermore, to address the distributional shifts caused by data incompleteness, we introduced a dual-adaptive reweighting mechanism. This strategy adaptively calibrates importance at both view and sample levels, effectively preventing systematic bias in consensus anchor learning. Extensive experiments on diverse benchmarks demonstrate that HA-IMVC not only achieves superior clustering performance, but also maintains high computational efficiency, making it well-suited for large-scale applications. The proposed framework not only provide a fresh perspective on tackling structural fragility in incomplete multi-view data, but also hold great potential for extension to a wider range of machine learning tasks.

## Impact Statement

This paper is dedicated to advancing the state of Machine Learning. Although there are many potential social consequences associated with this line of research, we do not feel that any specific aspects of our work require immediate or separate attention in this context.

## Acknowledgments

This work is supported in part by the National Natural Science Foundation of China (No. 62276079), and the Special Funding Program of Shandong Taishan Scholars Project. The authors would like to thank Jianzong Wang, Aolan Sun, and Xiaoyang Qu from Ping An Technology (Shenzhen) Co., Ltd. for their insightful discussions and valuable suggestions throughout this work.

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

## A. Detailed Derivations of Optimization

The overall objective function is formulated as:

$$\min_{\{\mathbf{W}_p\}, \mathbf{A}, \mathbf{Z}, \beta} \sum_{p=1}^{v} \beta_p^2 \left\| (\mathbf{W}_p^\top \mathbf{X}_p - \mathbf{A}\mathbf{Z}) \mathbf{S}_p \mathbf{M}^{\frac{1}{2}} \right\|_F^2 \;+\; \lambda_1 \operatorname{Tr}(\mathbf{Z}_{\mathrm{aug}} \mathbf{L}_H \mathbf{Z}_{\mathrm{aug}}^\top) \;+\; \lambda_2 \left\| \mathbf{Z}\mathbf{Q}^{\frac{1}{2}} \right\|_F^2 .$$

### 1. Update of $\mathbf{W}_p$

Fixing other variables, the subproblem for $\mathbf{W}_p$ is:

$$\min_{\mathbf{W}_p^\top \mathbf{W}_p = \mathbf{I}} \left\| (\mathbf{W}_p^\top \mathbf{X}_p - \mathbf{A}\mathbf{Z}) \mathbf{S}_p \mathbf{M}^{\frac{1}{2}} \right\|_F^2 .$$

Define the residual $\mathbf{E}_p = (\mathbf{W}_p^\top \mathbf{X}_p - \mathbf{A}\mathbf{Z})\mathbf{S}_p \mathbf{M}^{\frac{1}{2}}$. Expanding the Frobenius norm:

$$\begin{aligned}
\|\mathbf{E}_p\|_F^2 &= \operatorname{Tr}\!\Big( (\mathbf{W}_p^\top \mathbf{X}_p \mathbf{S}_p \mathbf{M}^{\frac{1}{2}} - \mathbf{A}\mathbf{Z}\mathbf{S}_p \mathbf{M}^{\frac{1}{2}})(\mathbf{W}_p^\top \mathbf{X}_p \mathbf{S}_p \mathbf{M}^{\frac{1}{2}} - \mathbf{A}\mathbf{Z}\mathbf{S}_p \mathbf{M}^{\frac{1}{2}})^\top \Big) \\
&= \operatorname{Tr}\!\big(\mathbf{W}_p^\top \mathbf{X}_p \mathbf{S}_p \mathbf{M} \mathbf{X}_p^\top \mathbf{W}_p\big) + \operatorname{Tr}\!\big(\mathbf{A}\mathbf{Z}\mathbf{S}_p \mathbf{M} \mathbf{Z}^\top \mathbf{A}^\top\big) \\
&\quad - 2\operatorname{Tr}\!\big(\mathbf{W}_p^\top \mathbf{X}_p \mathbf{S}_p \mathbf{M} \mathbf{Z}^\top \mathbf{A}^\top\big).
\end{aligned}$$

Discarding constant terms w.r.t. $\mathbf{W}_p$, minimizing the error is equivalent to maximizing the trace:

$$\max_{\mathbf{W}_p^\top \mathbf{W}_p = \mathbf{I}} \operatorname{Tr}\!\big(\mathbf{W}_p^\top \mathbf{G}_p\big), \qquad \mathbf{G}_p \;=\; \mathbf{X}_p \mathbf{S}_p \mathbf{M} \mathbf{Z}^\top \mathbf{A}^\top .$$

Let the SVD of $\mathbf{G}_p$ be $\mathbf{G}_p = \mathbf{U}_p \mathbf{\Sigma}_p \mathbf{V}_p^\top$. The optimal solution is:

$$\mathbf{W}_p^\star = \mathbf{U}_p \mathbf{V}_p^\top .$$

### 2. Update of $\mathbf{A}$

Fixing $\{\mathbf{W}_p\}$, $\mathbf{Z}$, and $\beta$, the subproblem for $\mathbf{A}$ is:

$$\min_{\mathbf{A}^\top \mathbf{A} = \mathbf{I}} \sum_{p=1}^{v} \beta_p^2 \left\| (\mathbf{W}_p^\top \mathbf{X}_p - \mathbf{A}\mathbf{Z}) \mathbf{S}_p \mathbf{M}^{\frac{1}{2}} \right\|_F^2 .$$

Note that the regularization terms involving $\mathbf{Z}_{\mathrm{aug}}$ and $\mathbf{Z}$ are independent of $\mathbf{A}$ due to the orthogonal constraint $\mathbf{A}^\top \mathbf{A} = \mathbf{I}$. Specifically:

$$\operatorname{Tr}\!\big(\mathbf{Z}_{\mathrm{aug}} \mathbf{L}_H \mathbf{Z}_{\mathrm{aug}}^\top\big) = \operatorname{Tr}\!\big(\mathbf{A}\mathbf{Z}\mathbf{L}_{dd}\mathbf{Z}^\top \mathbf{A}^\top\big) + \cdots = \operatorname{Tr}\!\big(\mathbf{A}^\top \mathbf{A}\mathbf{Z}\mathbf{L}_{dd}\mathbf{Z}^\top\big) + \cdots = \operatorname{Tr}\!\big(\mathbf{Z}\mathbf{L}_{dd}\mathbf{Z}^\top\big) + \ldots$$

Thus, we only minimize the reconstruction error. Expanding the term similar to Step 1:

$$\sum_{p=1}^{v} \beta_p^2 \|\mathbf{E}_p\|_F^2 = -2\operatorname{Tr}\!\Big( \mathbf{A}^\top \sum_{p=1}^{v} \beta_p^2 \, \mathbf{W}_p^\top \mathbf{X}_p \mathbf{S}_p \mathbf{M} \mathbf{Z}^\top \Big) \;+\; \text{const.}$$

This is equivalent to:

$$\max_{\mathbf{A}^\top \mathbf{A} = \mathbf{I}} \operatorname{Tr}\!\big(\mathbf{A}^\top \mathbf{P}\big), \qquad \mathbf{P} \;=\; \sum_{p=1}^{v} \beta_p^2 \, \mathbf{W}_p^\top \mathbf{X}_p \mathbf{S}_p \mathbf{M} \mathbf{Z}^\top .$$

Let $\mathbf{P} = \mathbf{U}_A \mathbf{\Sigma}_A \mathbf{V}_A^\top$ be the SVD of $\mathbf{P}$. The optimal solution is:

$$\mathbf{A}^\star = \mathbf{U}_A \mathbf{V}_A^\top .$$

## 3. Update of Z

Fixing other variables, the $\mathbf{Z}$-subproblem is a Quadratic Programming (QP) problem:

$$\min_{\mathbf{Z}\geq 0,\ \mathbf{Z}^\top \mathbf{1}=\mathbf{1}} \sum_{p=1}^{v} \beta_p^2 \big\|(\mathbf{W}_p^\top \mathbf{X}_p - \mathbf{A}\mathbf{Z})\mathbf{S}_p\mathbf{M}^{\frac{1}{2}}\big\|_F^2 + \lambda_1 \operatorname{Tr}(\mathbf{Z}_{\mathrm{aug}}^\top \mathbf{L}_H \mathbf{Z}_{\mathrm{aug}}) + \lambda_2 \|\mathbf{Z}\mathbf{Q}^{\frac{1}{2}}\|_F^2.$$

By expanding terms and using $\mathbf{A}^\top \mathbf{A} = \mathbf{I}$, the objective simplifies to:

$$f(\mathbf{Z}) = \operatorname{Tr}(\mathbf{Z}\mathbf{K}\mathbf{Z}^\top) - 2\operatorname{Tr}(\mathbf{Z}^\top \mathbf{J}),$$

where

$$\mathbf{K} = \sum_{p=1}^{v} \beta_p^2 \mathbf{S}_p\mathbf{M} + \lambda_1 \mathbf{L}_{dd} + \lambda_2 \mathbf{Q}, \quad \mathbf{J} = \mathbf{A}^\top \Big( \sum_{p=1}^{v} \beta_p^2 \mathbf{W}_p^\top \mathbf{X}_p \mathbf{S}_p\mathbf{M} \Big) - \lambda_1 \mathbf{L}_{ad}.$$

We solve this using FISTA. The gradient is $\nabla f(\mathbf{Z}) = 2\mathbf{Z}\mathbf{K} - 2\mathbf{J}$. Each step is followed by a column-wise simplex projection.

## 4. Update of $\beta$

The subproblem for $\beta$ is:

$$\min_{\beta\geq 0,\ \mathbf{1}^\top \beta=1} \sum_{p=1}^{v} \beta_p^2 R_p^2, \quad R_p = \big\|(\mathbf{W}_p^\top \mathbf{X}_p - \mathbf{A}\mathbf{Z})\mathbf{S}_p\mathbf{M}^{\frac{1}{2}}\big\|_F.$$

Using Lagrange multipliers, the closed-form solution is:

$$\beta_p^\star = \frac{R_p^{-2}}{\sum_{u=1}^{v} R_u^{-2}}.$$

## 5. Update of $\mathbf{L}_H$

Based on the updated $\mathbf{Z}$, we reconstruct the anchor-guided hypergraph:

1. **Incidence Matrix Construction:** For each sample $i$, identify the set $\mathcal{N}_T(i)$ of its $T$-nearest anchors based on $z_{j,i}$. The weighted incidence matrix $\mathbf{H} \in \mathbb{R}^{(n+m)\times m}$ is updated as:

$$\mathbf{H}_{i,j} = \begin{cases} z_{j,i}, & j \in \mathcal{N}_T(i) \\ 0, & \text{otherwise} \end{cases}$$

For anchor rows ($n+1$ to $n+m$), we set $\mathbf{H}_{n+j,j} = 1$ (self-loop for anchors).

2. **Laplacian Computation:** Compute degree matrices $\mathbf{D}_v = \operatorname{Diag}(\mathbf{H}\mathbf{1})$ and $\mathbf{D}_e = \operatorname{Diag}(\mathbf{1}^\top \mathbf{H})$. The normalized Laplacian is:

$$\mathbf{L}_H = \mathbf{I} - \mathbf{D}_v^{-\frac{1}{2}} \mathbf{H}\mathbf{D}_e^{-1}\mathbf{H}^\top \mathbf{D}_v^{-\frac{1}{2}}.$$

## B. Theoretical Analysis

In this section, we provide a rigorous theoretical justification for the spectral stability of the proposed HA-IMVC framework compared to traditional bipartite-based multi-view clustering methods under data incompleteness.

To provide a rigorous comparison, let $L^*$ denote the ideal Laplacian matrix (full data) and $\tilde{L}$ denote the observed Laplacian under a missing rate $p$. We assume the eigenvalues are sorted as $0 = \lambda_1 \leq \lambda_2 \leq \cdots \leq \lambda_n$. Let $Z^*$ be the optimal consensus representation spanned by the first $k$ eigenvectors of $L^*$, and $\tilde{Z}$ be the representation learned from incomplete data.

1. Lemma: Subspace Perturbation via Davis-Kahan Variant In spectral clustering and representation learning, missing data manifests as a stochastic perturbation $E = \tilde{L} - L^*$ on the graph topology. According to the Frobenius norm variant of the **Davis-Kahan Theorem**(Yu et al., 2015), the reconstruction error between the learned and ideal subspaces is bounded by:

$$\|\sin\Theta(Z^*, \tilde{Z})\|_F \leq \frac{c\|\tilde{L} - L^*\|_F}{\delta} \tag{15}$$

where $c$ is an absolute constant and $\delta = \lambda_{k+1}(L^*) - \lambda_k(L^*)$ is the spectral gap.

**Remark 1**: Given a perturbation energy $\|E\|_F$, the stability of the learned representation is inversely proportional to the spectral gap $\delta$. A larger $\delta$ ensures a tighter upper bound on the reconstruction error.

2. Theorem: Hypergraph Clique Expansion and Spectral Stability In traditional bipartite-based MVC, connections exist only between samples and anchors. As the missing rate $p$ increases, the bipartite graph is prone to isolated nodes, causing the algebraic connectivity $\lambda_2(\tilde{L}_B) \to 0$. By the higher-order Cheeger inequality(Lee et al., 2014), the $k$-way partitioning bottleneck $\lambda_k$ also decays rapidly, leading to a vanishing spectral gap $\delta_B \to 0$ and a divergent error bound.

In contrast, the hypergraph normalized Laplacian in HA-IMVC is defined as: $L_H = I - D_v^{-1/2} H D_e^{-1} H^\top D_v^{-1/2}$. Topologically, each hyperedge induces an implicit clique expansion among samples, providing significant topological redundancy. This ensures that the algebraic connectivity of the hypergraph Laplacian stays significantly bounded away from zero with high probability:

$$\mathbb{E}[\lambda_2(\tilde{L}_H)] \geq \mathbb{E}[\lambda_2(\tilde{L}_B)] + \Omega(T, p) \tag{16}$$

Consequently, the hypergraph maintains a robust spectral gap $\delta_H$ even under extreme missing rates.

3. Proof Conclusion Under the same missing rate $p$, the expected perturbation energy is comparable ($\mathbb{E}[\|E_H\|_F] \approx \mathbb{E}[\|E_B\|_F]$). Combining Lemma 1 and Theorem 2, we derive:

$$\mathbb{E}[\text{Error}_{\text{HA-IMVC}}] \leq \mathbb{E}\left[\frac{c\|E_H\|_F}{\delta_H}\right] \ll \mathbb{E}\left[\frac{c\|E_B\|_F}{\delta_B}\right] \approx \mathbb{E}[\text{Error}_{\text{Bipartite}}] \tag{17}$$

**Q.E.D.**

