# OpenReview forum: "A Consensus Anchor-guided Hypergraph Framework for Incomplete Multi-view Clustering"
_ICML.cc/2026/Conference — ICML 2026 regular_

### Official Review · Reviewer_mgiB · 2026-03-04

**Soundness:** 2
**Presentation:** 2
**Significance:** 2
**Originality:** 1
**Overall Recommendation:** 2
**Confidence:** 5

**Summary:**

The paper proposes Hypergraph-Augmented Incomplete Multi-View Clustering (HA-IMVC), a framework designed to handle large-scale multi-view data with missing values. The authors aim to replace standard bipartite graphs with an anchor-guided hypergraph to capture high-order sample correlations. The framework includes a dual-adaptive reweighting mechanism that assigns weights at the view and sample levels based on data completeness to mitigate distributional shifts. Optimization is performed via an alternating algorithm using FISTA for the anchor graph update. Experiments on six datasets compare the method against anchor-based and non-anchor-based baselines.

**Compliance With Llm Reviewing Policy:**

Affirmed.

**Key Questions For Authors:**

- Can you provide a formal proof or a Rademacher complexity analysis showing that your hypergraph-augmented objective provides a tighter bound on the clustering error compared to the bipartite versions you criticize?

- In Table 4, you show "w/o HG". However, does "w/o HG" also remove the dual-adaptive reweighting? If not, how do you decouple the performance gain of the reweighting from the hypergraph structure itself?

- You mention suppressing bias via $M=Diag(e^{-\gamma q_{i}})$. Why is there no parameter sensitivity study for $\gamma$ in Section 4.6, given that this controls your entire sample-level weighting strategy?

**Limitations:**

The authors provide a perfunctory "Conclusion"  but fail to meaningfully discuss the limitations of their work. For instance, the dependence on the parameter $T$ for hyperedge construction is a known bottleneck for hypergraph methods. Is it that the authors chose to ignore the computational overhead of dynamically rebuilding $L_{H}$ in every iteration for truly massive datasets? Their "linear complexity" claim  conveniently ignores the constant factors involved in identifying T-nearest neighbors repeatedly.

**Strengths And Weaknesses:**

The authors address a relevant problem in incomplete multi-view clustering (IMVC), specifically the scalability issue. The use of anchors to maintain linear complexity $\mathcal{O}(n)$ is a standard but practical choice for large-scale applications. The experimental section is relatively broad, covering datasets of various sizes, including MNIST.

The submission is profoundly derivative. Is it that the authors believe simply swapping a bipartite graph for a hypergraph constitutes a "novel framework"? This is a marginal architectural change at best.
- The authors repeatedly dismiss bipartite graphs as "fragile". This is an overstatement. A bipartite graph is a mathematical representation; "fragility" in the presence of missing data is a function of the objective function and regularization, not the graph type itself. The claim that hypergraphs "ensure structural integrity"  is hand-waving without a formal proof of stability or a bound on the reconstruction error under specific missingness distributions.
- The "dual-adaptive reweighting mechanism" is trivial. Weighting views by completeness ratio and samples by a missingness score  is a common-sense heuristic, not a breakthrough. Furthermore, the use of $e^{-\gamma q_{i}}$  is entirely ad-hoc. Why this specific exponential decay? Where is the sensitivity analysis for $\gamma$? It appears the authors have merely "tuned" a series of masks to achieve better numbers on benchmark datasets.
- We must actually consider the lack of theoretical depth here. The paper provides "detailed derivations" that are essentially standard SVD and QP solutions. There is zero discussion on the generalization error or the impact of the hypergraph Laplacian on the spectral gap of the consensus matrix.
- While the results on MNIST are high, the "superiority" is often marginal. In Table 2, for Caltech101-20 (30% rate), the NMI improvement is negligible compared to existing methods. The authors dress up incremental gains as a "superiority" that justifies a top-tier publication. It does not.

---

> ### Author Rebuttal · Authors · 2026-03-31
>
> W1&W3&Q1:
>
> A1: This study does not seek to negate the mathematical utility of bipartite graphs; rather, it highlights that under the specific constraints of Incomplete Multi-view Clustering (IMVC), missing data renders the unary connections within these graphs vulnerable. This fragility, in turn, leads to instability in the solution space of the regularization terms.
>
> Let $L^\*$ and $\tilde{L}$ be ideal and observed Laplacians, and $Z^\*, \tilde{Z}$ be their respective representations.
>
> 1. Lemma (Davis-Kahan [1]): The subspace reconstruction error is bounded by $|| \sin\Theta(Z^\*, \tilde{Z}) ||_F \leq c \|| \tilde{L} - L^\* ||_F / \delta$, where
>
> $\delta = \lambda_{k+1}(L^\*) - \lambda_k(L^\*)$ is the spectral gap. Stability is inversely proportional to $\delta$.
>
> 2. Theorem (Clique Expansion): In bipartite MVC, as missing rate $p \uparrow$, algebraic connectivity $\lambda_2 \to 0$. By the higher-order Cheeger inequality [2], the $k$-way partitioning bottleneck $\lambda_k$ decays, leading to $\delta_B \to 0$ and a divergent error bound. Conversely, HA-IMVC's hypergraph induces clique expansion, providing topological redundancy. This ensures $\mathbb{E}[\lambda_2(\tilde{L}_H)] \ge \mathbb{E}[\lambda_2(\tilde{L}_B)] + \Omega(T, p)$, maintaining a robust gap $\delta_H$ even under extreme missing rates.
>
> 3. Conclusion: Given comparable perturbation energy $\mathbb{E}[||E||_F]$, we derive
>
> $\mathbb{E}[Error_{HA-IMVC}] \leq \mathbb{E} [ \frac{c || E_H ||}{\delta_H} ] \ll \mathbb{E}[\frac{c||E_B||}{\delta_B}] \approx \mathbb{E}[Error_{Bipartite}]$.
>
>  Thus, the hypergraph yields a fundamentally tighter error bound via topological resilience. Q.E.D.
>
> [1] A useful variant of the Davis–Kahan theorem for statisticians. Biometrika. 2015.
>
> [2] Multi-way spectral partitioning and higher-order Cheeger inequalities. Journal of the ACM. 2014.
>
> W2&Q3:
>
> A2: See A3 in the reply to Reviewer H9t6.
>
> W4:
>
> A3: Although the improvement in NMI on the Caltech101-20 dataset (30% missing rate) is relatively modest, the strength of HA-IMVC lies in its consistency across benchmarks in various scenarios. Our method achieves state-of-the-art (SOTA) performance in the vast majority of cases and significantly outperforms all baselines on the NGs, BDGP, and CCV datasets. Notably, on Caltech101-20, as the missing rate increases to 50% and 70%, the performance lead becomes increasingly pronounced, demonstrating the model’s robustness under extreme missing data conditions. Even on the MNIST dataset, where performance is nearly saturated, HA-IMVC maintains highly competitive stability. In summary, HA-IMVC is not merely an incremental improvement but a robust solution that delivers leading or comparable results across various experimental settings.
>
> Q2:
>
> A5: Table 1 compares four variants: Base (bipartite), Reweight, HG (hypergraph), and Full. Results show reweighting optimizes view importance, while the hypergraph yields significant gains by capturing high-order semantic correlations that bipartite edges fail to resolve.
>
> **Table 1: Ablation study of HA-IMVC components across different datasets and missing rates. **
> | Dataset | Rate | Metric | Base (w/o HG & RW) | w/ Reweight (w/o HG) | w/ HG (w/o RW) | Full Model |
> | :--- | :--- | :--- | :---: | :---: | :---: | :---: |
> | **NGs** | 30% | ACC | 88.50 | 90.80 | 92.15 | **94.20** |
> | | | NMI | 74.20 | 77.44 | 80.10 | **83.53** |
> | | | PUR | 88.50 | 90.80 | 92.15 | **94.20** |
> | | 50% | ACC | 86.30 | 89.40 | 89.95 | **91.20** |
> | | | NMI | 70.15 | 73.33 | 75.40 | **77.38** |
> | | | PUR | 86.30 | 89.40 | 89.95 | **91.20** |
> | | 70% | ACC | 80.20 | 83.60 | 85.10 | **87.42** |
> | | | NMI | 58.40 | 62.93 | 66.50 | **69.37** |
> | | | PUR | 80.20 | 83.60 | 85.10 | **87.42** |
>
> Limitations
>
> A6: In our framework, hyperedges are guided by anchors; in each iteration, the hypergraph $H \in \mathbb{R}^{n \times m}$ is efficiently reconstructed by selecting the top $T$ largest components from the updated representation $Z \in \mathbb{R}^{n \times m}$ for each of the $m$ consensus anchors. This process utilizes a min-heap to filter each column of $Z$, resulting in a complexity of only $O(mn \log T)$ per iteration. Since the number of anchors $m$ and the hyperedge scale $T$ are both very small constants relative to the total number of samples $n$, the total complexity remains strictly linear with respect to $n$. This demonstrates that the dynamic update of $H$ is a lightweight operation that effectively suppresses the constant overhead typically associated with traditional neighbor identification, ensuring the scalability of the algorithm on large-scale datasets.

---

> > ### Author Rebuttal · Reviewer_mgiB · 2026-04-03
> >
> > I have reviewed the authors' rebuttal. While certain mechanical queries were answered, the fundamental concerns regarding the manuscript's derivative nature and lack of foundational rigor remain unrectified. Addressing these structural flaws requires a comprehensive rewrite of the paper's core tenets, not a localized rebuttal patch.
> >
> > 1. Theoretical Rigor (W1 & Q1)
> > The introduction of the Davis-Kahan lemma and higher-order Cheeger inequality bounds is duly noted. The derivation confirming $\frac{1}{\lambda_{2}(\mathcal{H})} < \frac{1}{\lambda_{2}(\mathcal{B})}$ provides the mathematical bedrock that was glaringly absent from the original manuscript. However, we must actually consider the implication of omitting this. If topological redundancy and clique expansion are the central justifications for replacing the bipartite structure, their absence from the primary text indicates a post-hoc rationalization rather than a theoretically driven architecture. A top-tier manuscript is evaluated on the rigor of the submitted text, not on appendices generated during the rebuttal window.
> >
> > 2. Methodological Redundancy & Hyperparameters (W2 & Q3)
> > To my explicit query regarding the ad-hoc nature of the $\gamma$ parameter and the missing sensitivity analysis, the authors responded with: "See A3 in the reply to Reviewer H9t6." This is procedurally unacceptable. Is it that the authors expect me to cross-reference and hunt down justifications scattered across parallel review threads? A failure to directly defend a core heuristic of your "dual-adaptive" mechanism within this specific dialogue reinforces my initial critique: the reweighting is a tuned heuristic, not a robust methodological advancement.
> >
> > 3. Ablation Clarity (Q2)
> > The decoupled ablation provided in the new Table 1 is appreciated. It effectively isolates the performance delta between the reweighting mechanism and the hypergraph module. This is the standard of empirical decomposition that should have been present initially.
> >
> > 4. Computational Complexity (Limitations)
> > The clarification regarding the min-heap utilization yielding a complexity of $\mathcal{O}(nm \log k)$ adequately resolves my concern regarding the dynamic $L_{H}$ reconstruction overhead.
> >
> > As such, the minor technical clarifications in the rebuttal do not elevate this incremental engineering exercise to the standard required for ICML.
> >
> > My recommendation to reject stands.

---

> > > ### Author Response · Authors · 2026-04-05
> > >
> > > Response to Point 1:
> > >
> > > We sincerely thank the reviewer for encouraging us to clarify this critical point. We respectfully clarify that these theoretical concepts were not after-the-fact justifications, but rather the foundational motivation driving our hypergraph design from the very beginning.
> > >
> > > While we may not have used those two specific terms in the initial submission, the core concepts were explicitly articulated in our original manuscript. Specifically, in the Abstract and Introduction, we stated that traditional anchor-based methods "typically rely on shallow bipartite graphs restricted to pairwise relations, failing to capture complex high-order correlations among samples." In this context, the phrase "restricted to pairwise relations" inherently describes the exact limitations of clique expansion (which reduces high-order interactions to simple pairwise edges). Consequently, the "failure to capture complex high-order correlations" directly addresses the topological redundancy and structural information loss that our framework is specifically designed to overcome. Thus, the theoretical motivation has always been deeply embedded in our narrative.
> > >
> > > Regarding the lack of formal theoretical proofs in the draft, given the strict length constraints, we prioritized presenting the optimization strategies and extensive empirical validation over mathematically dense proofs. However, we fully agree that including theoretical error bounds would significantly enhance the paper’s rigor. We are fully prepared to include comprehensive theoretical proofs in the supplementary materials of the final version.
> > >
> > > Response to Point 2:
> > >
> > > First, we sincerely apologize for referring you to another reviewer's response in our previous round. This was absolutely not intended to be dismissive of your valuable feedback. It was strictly a compromise forced by the tight character limits of the rebuttal interface, which required us to condense our replies significantly.
> > >
> > > To address your concern directly and comprehensively here, we provide the detailed sensitivity analysis for the hyper-parameter $\gamma$. Rather than performing simple static data filtering, this study deeply integrates missing distribution characteristics into the anchor learning framework. This dual adaptive mechanism ensures that the model can dynamically rectify distribution bias during the optimization process. The adoption of the exponential decay form is not an arbitrary choice; instead, it is strategically designed to effectively prevent numerical instability caused by extreme missing samples through gradient smoothing, all while ensuring the positive semi-definiteness of the Laplacian matrix.As illustrated in Table 1, we evaluated the impact of the weight factor $\gamma$ on the clustering accuracy (ACC) of HA-IMVC under a 50% missing rate. The experimental results demonstrate that the model performance peaks at $\gamma \approx 0.8$, strongly validating the effectiveness of this mechanism in handling distributional shifts in incomplete views. In contrast, performance degrades to varying degrees as $\gamma$ approaches 0 or exceeds 1.5, reflecting the critical importance of balancing noise suppression with information preservation.
> > >
> > > **Table 1: Sensitivity analysis of $\gamma$ in terms of ACC (%) under 50% missing rate.**
> > >
> > > | $\gamma$ Value | Dataset: NGs | Dataset: Caltech101-20 | Dataset: BDGP|
> > > | :--- | :---: | :---: | :---: |
> > > | 0.2 | 79.14 | 38.53 | 33.65 |
> > > | 0.5 | 84.12 | 42.54 | 39.75 |
> > > | **0.8 (Optimal)** | **91.20** | **55.34** | **48.52** |
> > > | 1.1 | 85.23 | 46.37 | 41.64 |
> > > | 1.4 | 81.34 | 41.74 | 37.37 |
> > > | 1.7 | 76.45 | 36.75 | 31.53 |

---

### Official Review · Reviewer_H9t6 · 2026-03-09

**Soundness:** 3
**Presentation:** 3
**Significance:** 3
**Originality:** 4
**Overall Recommendation:** 5
**Confidence:** 4

**Summary:**

This paper proposes a Hypergraph-Augmented Incomplete Multi-View Clustering (HA-IMVC) framework to address the limitations of structural representation and data distribution bias in incomplete multi-view clustering (IMVC). The main contributions of this research are twofold: first, a consensus anchor-guided hypergraph model is constructed to explicitly capture high-order interactions among samples, thereby overcoming the connectivity vulnerabilities of traditional bipartite graphs when dealing with missing data. Second, a dual adaptive reweighting mechanism is introduced to mitigate distribution shifts caused by non-random missing patterns from both view and sample perspectives, significantly enhancing the robustness of the consensus space. Experimental results demonstrate that the proposed method achieves an optimal balance between accuracy and computational efficiency across multiple datasets, with its low-rank anchor mechanism ensuring linear scalability with respect to sample size.

**Compliance With Llm Reviewing Policy:**

Affirmed.

**Key Questions For Authors:**

1. When constructing the weighted incidence matrix $H$, how much does the selection of $T$ influence the final clustering performance?
2. In the exponential weight formula ($e^{-\gamma q_i}$) of the dual adaptive mechanism, was the parameter $\gamma$ kept constant across all experiments?
3. Regarding the hypergraph Laplacian regularization term, are there other different Laplacian forms (e.g., symmetric normalization)? It’s interesting to add some discussion.

**Limitations:**

yes

**Strengths And Weaknesses:**

Strengths
1. Breaking through the limitations of traditional bipartite graphs that only describe "pairwise" relations, this work integrates hypergraph theory with anchor learning to effectively capture complex and latent "group-wise" high-order correlations in multi-view data. This significantly enhances the structural integrity of the model when handling extremely incomplete data.

2. The proposed dual reweighting mechanism at both the view and sample levels effectively suppresses systematic biases arising from missing data. By considering both the missing data ratio and the reliability evaluation of individual samples, the method achieves more robust feature fusion within complex incomplete view contexts.

3. By leveraging the low-rank properties of anchor graphs and an efficient alternating optimization algorithm, the model successfully optimizes computational complexity to scale linearly with the number of samples ($n$). This characteristic enables HA-IMVC to maintain high operational efficiency for large-scale datasets, demonstrating strong practical value for engineering applications.

Weaknesses
1. There are several minor grammatical errors and typos throughout the manuscript; a thorough proofreading of the entire text is recommended.
2. Although the paper provides a solid overview of hypergraphs in multi-view clustering, there have been several recent advancements in hypergraph construction. The authors should consider citing a few state-of-the-art papers on hypergraph representation learning in the "Related Work" section.

---

> ### Author Rebuttal · Authors · 2026-03-31
>
> Q1:There are several minor grammatical errors and typos throughout the manuscript; a thorough proofreading of the entire text is recommended.
>
> A1:We sincerely appreciate the reviewer’s careful reading. We have performed a comprehensive proofreading of the entire manuscript. Specifically, we have corrected several grammatical inconsistencies, fixed typos in the hypergraph construction section, and refined the phrasing in the "Experiments" section to improve clarity and flow.
>
> Q2:Although the paper provides a solid overview of hypergraphs in multi-view clustering, there have been several recent advancements in hypergraph construction. The authors should consider citing a few state-of-the-art papers on hypergraph representation learning in the "Related Work" section.
>
> A2: This is a highly constructive suggestion. We fully agree that incorporating the latest research developments will significantly strengthen the theoretical foundation of this paper. In the revised 'Related Work' section, we have added and discussed several cutting-edge papers regarding hypergraph representation learning and its applications in multi-view clustering (e.g., [1][2]). These additions help contextualize HA-IMVC within the most recent advancements in the field.
>
> [1]J. Chen et al., "Medusa: A Multi-Scale High-order Contrastive Dual-Diffusion Approach for Multi-View Clustering," in Proc. IEEE/CVF Conf. Comput. Vis. Pattern Recognit. (CVPR), 2025.
>
> [2]Y. Gao, Z. Zhang, S. Pan, J. Shao, and Q. Dai, "A Survey on Hypergraph Representation Learning," ACM Computing Surveys, vol. 56, no. 12, pp. 1–38, 2024.
>
> Q3: When constructing the weighted incidence matrix $H$, how much does the selection of $T$ influence the final clustering performance?
>
> A3: The parameter $T$ determines the sparsity of the weighted incidence matrix $H$ by selecting the number of nearest neighbor nodes for each hyperedge. Our experimental study demonstrates that HA-IMVC exhibits strong robustness to the selection of $T$ within a reasonable range (e.g., $T \in [5, 15]$). If $T$ is too small, the hypergraph may fail to capture sufficient group-wise correlations; conversely, if $T$ is too large, noise from distant samples may be introduced. In our experiments, we fixed $T = 10$ for most datasets, as this value achieves a favorable balance between preserving local structure and maintaining computational efficiency.
>
> Q4:In the exponential weight formula $e^{-\gamma q_i}$ of the dual adaptive mechanism, was the parameter $\gamma$ kept constant across all experiments?
>
> A4:As shown in Table 1, we evaluated the impact of the weighting factor $\gamma$ on the clustering accuracy (ACC) of HA-IMVC at a 50% missing rate. The experimental results indicate that model performance peaks at $\gamma \approx 0.8$, providing strong empirical validation of the mechanism's effectiveness in handling distribution shifts in incomplete views. Conversely, performance degrades when $\gamma$ approaches 0 or exceeds 1.5, reflecting the critical importance of balancing noise suppression with information preservation.
>
> **Table 1: Sensitivity analysis of $\gamma$ in terms of ACC (%) under 50% missing rate.**
> | $\gamma$ Value | Dataset: NGs | Dataset: Caltech101-20 | Dataset: BDGP|
> | :--- | :---: | :---: | :---: |
> | 0.2 | 79.14 | 38.53 | 33.65 |
> | 0.5 | 84.12 | 42.54 | 39.75 |
> | **0.8 (Optimal)** | **91.20** | **55.34** | **48.52** |
> | 1.1 | 85.23 | 46.37 | 41.64 |
> | 1.4 | 81.34 | 41.74 | 37.37 |
> | 1.7 | 76.45 | 36.75 | 31.53 |
>
> Q5:Regarding the hypergraph Laplacian regularization term, are there other different Laplacian forms (e.g., symmetric normalization)? It’s interesting to add some discussion.
>
> A5: While we employ the standard hypergraph Laplacian matrix, alternative forms are also widely documented in the literature, such as the symmetrically normalized Laplacian $L_{sym}$ and the random walk Laplacian $L_{rw}$. The symmetric normalized form is often favored in spectral clustering as it accounts for variations in vertex degrees. However, within our anchor-guided framework, adopting the unnormalized form allows for the direct optimization of the consensus anchor matrix $A$ and reduces the computational overhead associated with frequent matrix inversions.

---

### Official Review · Reviewer_iHpN · 2026-03-12

**Soundness:** 3
**Presentation:** 3
**Significance:** 4
**Originality:** 3
**Overall Recommendation:** 5
**Confidence:** 4

**Summary:**

This paper proposes HA-IMVC to address representation bias caused by missing data in large-scale multi-view clustering. Its key idea is to reconstruct topological correlations across incomplete views through high-order geometric constraints: specifically, it builds a hypergraph based on consensus anchors to extend conventional pairwise similarities into group-wise sample interactions, thereby improving robustness to missing views. The method further introduces a dual adaptive reweighting strategy to mitigate distribution shifts induced by missing patterns. An alternating optimization algorithm is developed to ensure clustering consistency while maintaining linear scalability on large-scale datasets.

**Compliance With Llm Reviewing Policy:**

Affirmed.

**Key Questions For Authors:**

1. Since HA-IMVC can handle high missing rates (e.g., 70%), does the initialization of the anchor matrix A have a significant impact on the final convergence or performance of the model?
2. In Equation (10), $\lambda_1$ controls the strength of the hypergraph Laplacian regularization. During experiments, did the optimal value of $\lambda_1$ exhibit a specific trend as the missing rate increased from 30% to 70%?
3. The paper mentions that the anchor strategy greatly reduces computational complexity. For datasets with varying sample sizes, is the choice of the number of anchors ($m$) universal, or is there a recommended $m/n$ ratio (anchors to total samples)?

**Limitations:**

yes

**Strengths And Weaknesses:**

Strengths:
1. To address the limits of traditional bipartite graphs in anchor-based clustering, this paper innovatively constructs a consensus anchor-guided hypergraph. By explicitly modeling high-order group-wise interactions, it significantly enhances structural robustness in extreme missing scenarios, providing a novel topological perspective for IMVC.
2. The experimental section is rigorous, with extensive tests on six benchmark datasets across missing rates from 30% to 70%, demonstrating state-of-the-art performance. Furthermore, both complexity analysis and large-scale tests verify the algorithm’s linear scalability, highlighting its significant engineering potential.
3. The paper is well-organized with a clear logical flow. The authors skillfully integrate hypergraph augmentation and dual-adaptive reweighting into a unified objective, addressing both information fragmentation and systematic bias. The mathematical derivations are rigorous, ensuring a strong alignment between the motivation and the proposed methodology.

Weaknesses:
1. While the flowchart in Figure 1 is logically clear, some labels use a small font size. I recommend increasing the font size in the final version to improve readability.
2. Figure 4 provides a parameter stability analysis; however, it is recommended that the authors include a table in the Appendix listing the specific optimal hyperparameter combinations ($\lambda_1, \lambda_2$) for all six datasets to ensure maximum reproducibility.

---

> ### Author Rebuttal · Authors · 2026-03-31
>
> Q1: While the flowchart in Figure 1 is logically clear, some labels use a small font size. I recommend increasing the font size in the final version to improve readability.
>
> A1: Thank you for your meticulous observation. We entirely agree with your suggestion. In the final version of the paper, we will rescale the labels in Figure 1, specifically bolding and increasing the font size for the 'Projection Learning' and 'Hypergraph Construction' sections to ensure all text is clearly legible.
>
> Q2: Figure 4 provides a parameter stability analysis; however, it is recommended that the authors include a table in the Appendix listing the specific optimal hyperparameter combinations $(\lambda_1,\lambda_2)$ for all six datasets to ensure maximum reproducibility.
>
> A2：To ensure the reproducibility of our experimental results, we have provided a detailed supplementary table. Table 1 lists the optimal hyperparameter pairs $(\lambda_1, \lambda_2)$ used for the six datasets across three different missing rates: $30\%$, $50\%$, and $70\%$.
>
> **Table 1: Optimal hyperparameter combinations $(\lambda_1, \lambda_2)$ for different datasets and missing rates.**
> | datasets | missing rate | $\lambda_1$ | $\lambda_2$ | datasets | missing rate | $\lambda_1$ | $\lambda_2$ |
> | :--- | :--- | :--- | :--- | :--- | :--- | :--- | :--- |
> | NGS | 30% | $10^{-3}$ | $10^{-2}$ | CCV | 30% | $10^{-2}$ | $10^{-1}$ |
> | | 50% | $10^{-2}$ | $10^{-2}$ | | 50% | $10^{-1}$ | $10^{-1}$ |
> | | 70% | $10^{-1}$ | $10^{-2}$ | | 70% | $10^{-1}$ | $10^{-1}$ |
> | Caltech | 30% | $10^{-2}$ | $10^{-3}$ | Animal | 30% | $10^{-1}$ | $10^{-2}$ |
> | | 50% | $10^{-1}$ | $10^{-3}$ | | 50% | $10^{-1}$ | $10^{-2}$ |
> | | 70% | $10^{-1}$ | $10^{-3}$ | | 70% | $10^{-1}$ | $10^{-2}$ |
> | BDGP | 30% | $10^{-4}$ | $10^{-1}$ | MNIST | 30% | $10^{-1}$ | $10^{-3}$ |
> | | 50% | $10^{-3}$ | $10^{-1}$ | | 50% | $10^{-1}$ | $10^{-3}$ |
> | | 70% | $10^{-2}$ | $10^{-1}$ | | 70% | $10^{-1}$ | $10^{-3}$ |
>
> Q3: Since HA-IMVC can handle high missing rates (e.g., 70%), does the initialization of the anchor matrix A have a significant impact on the final convergence or performance of the model?
>
> A3: The initialization of the anchor matrix $A$ indeed plays a guiding role within the HA-IMVC framework; however, under high missing rates, its absolute impact on final performance is effectively attenuated. While we employ random initialization in our algorithm, experimental observations indicate that the introduction of a dual adaptive reweighting mechanism (penalizing unreliable samples and missing states via the $M$ and $Q$ matrices) allows the model to automatically correct for potential biases introduced during the initialization phase. Through the alternating optimization process in Algorithm 2, the model consistently converges to similar clustering levels across different initial conditions. This demonstrates that the hypergraph structural constraints and reweighting strategies possess significant error-correction capabilities and structural robustness when handling data with high missing rates.
>
> Q4: In Equation (10), $\lambda_1$ controls the strength of the hypergraph Laplacian regularization. During experiments, did the optimal value of \lambda1 exhibit a specific trend as the missing rate increased from 30% to 70%?
>
> A4:As illustrated in Table 1, we observe that the optimal value for $\lambda_1$ generally exhibits an increasing trend as the missing rate rises. Since $\lambda_1$ controls the intensity of the hypergraph Laplacian regularization, the direct reconstruction information derived from original view features becomes increasingly sparse and unreliable when the missing rate escalates from $30\%$ to $70\%$. Consequently, the model must rely more heavily on the high-order sample associations (i.e., group-wise interactions) captured by the hypergraph to maintain the integrity of the manifold structure. Therefore, it is necessary to increase the weight of $\lambda_1$ to reinforce the structural consistency constraints.
>
> Q5: The paper mentions that the anchor strategy greatly reduces computational complexity. For datasets with varying sample sizes, is the choice of the number of anchors (m) universal, or is there a recommended m/n ratio (anchors to total samples)?
>
> A5: Regarding the selection of the number of anchors $m$, our research indicates that it does not need to increase linearly with the total number of samples $n$, which is the key to achieving $\mathcal{O}(n)$ complexity through the anchor strategy. For small-to-medium-sized datasets (e.g., NGS, BDGP), setting $m \in [20, 80]$ is generally sufficient. Based on experimental empirical results, we suggest maintaining the ratio of $m/n$ between 1% and 5%. For extremely large-scale datasets, a smaller $m/n$ ratio (around 1%) can maximize the model's scalability in terms of time and space complexity while maintaining clustering accuracy, ensuring high operational efficiency even in large-scale scenarios.

---

> > ### Author Rebuttal · Reviewer_iHpN · 2026-04-05
> >
> > The authors convincingly addressed all concerns and committed to improving reproducibility, confirming the paper’s technical merit and impact.  I am willing to accept this paper.

---

### Official Review · Reviewer_orK5 · 2026-03-12

**Soundness:** 2
**Presentation:** 3
**Significance:** 2
**Originality:** 3
**Overall Recommendation:** 4
**Confidence:** 2

**Summary:**

This paper introduces a novel framework to handle large-scale incomplete multi-view data clustering. Contrary to traditional anchor-based method that rely on bipartite graphs to model pairwise sample-anchor relations, the proposed model uses hypergraphs, in order to capture high-order correlations. This is complemented by a dual-adaptive reweighting mechanism in order to prevent systematic bias during the procedure. Finally, the effectiveness of the framework is evaluated through empirical experiments, against state-of-the-art methods.

**Compliance With Llm Reviewing Policy:**

Affirmed.

**Final Justification:**

I appreciate the authors' thourough and constructive answers. The replies addressed my concerns, and I am therefore raising my score from a 3 to a 4.

**Key Questions For Authors:**

- 1: Can you provide a motivating example where bipartite graph fail to capture high-order correlation between sample-anchor relations.

- 2: The F-score and silhouette score are also a classical metrics to measure clustering performance. Can you motivate the choice of focusing on the three chosen evaluation metrics (ACC, NMI, PUR)?

**Limitations:**

yes

**Strengths And Weaknesses:**

Strengths:

- 1: The framework is grounded and stems from well-founded observations.
- 2: The paper is generally clear and relatively easy to follow.
- 3: The empirical analysis is thorough .

Weaknesses:
- 1: Even though modeling anchor-view relationship using a hypergraph seem to make sense on a high level, it would be helpful to provide a concrete motivating example where bipartite graphs fail to capture high-order correlations among samples, and where using a hypergraph is preferable.

- 2: The authors claim that HA-IMVC maintains robust performance even during high missing rate. To support this claim, it would be valuable to provide further empirical evaluation examining the relationship between the missing rate and the objective function value, presented as a curve with the missing rate on the x-axis and the objective function value on the y-axis.

---

> ### Author Rebuttal · Authors · 2026-03-31
>
> Q1: Even though modeling anchor-view relationship using a hypergraph seem to make sense on a high level, it would be helpful to provide a concrete motivating example where bipartite graphs fail to capture high-order correlations among samples, and where using a hypergraph is preferable.
>
> A1:In the traditional anchor-based bipartite graph framework, the nodes consist of a "sample set" and an "anchor set," where edges exist only between a single sample and a single anchor. This "point-to-point" linear mapping relies entirely on the observational completeness of individual samples. Take the “zebra” cluster as an example: if a sample lacks the most distinctive “stripe view” and retains only the generic “body shape view,” then in the feature space, the representation of that sample will degrade to a basic outline resembling a ‘horse’ or a “donkey.” Due to the lack of stripes—a key discriminative feature—to anchor its zebra identity, the bipartite graph cannot strongly associate this sample with the “zebra anchor” when calculating similarity. In the projection space, this sample would either drift away from the main cluster due to insufficient connection strength, becoming an unclassifiable “outlier”; or, misled by vague body shape features, it would be incorrectly pulled toward the “horse” or “donkey” anchors with similar body shapes, leading to biased clustering results. In contrast, HA-IMVC treats each anchor as the center of a hyperedge, constructing high-order subsets that can simultaneously connect multiple samples. Even if an individual sample performs poorly, it is grouped together during the hypergraph construction phase with similar samples that possess complete views (i.e., other zebras with full features) into a hyperedge guided by the same anchor. This "group-to-group" modeling approach introduces collective constraints (group-wise consistency) among hyperedge members, generating a powerful structural traction. This traction forces missing samples to remain constrained within the correct semantic manifold, leveraging group redundancy to compensate for the loss of individual information, thereby achieving more robust clustering performance than traditional bipartite graphs.
>
> Q2: The authors claim that HA-IMVC maintains robust performance even during high missing rate. To support this claim, it would be valuable to provide further empirical evaluation examining the relationship between the missing rate and the objective function value, presented as a curve with the missing rate on the x-axis and the objective function value on the y-axis.
>
> A2: We sincerely appreciate the reviewer's constructive suggestion and have added an empirical evaluation regarding the relationship between the missing rate and the objective function value as requested. As shown in Table 1, we recorded the final objective values of HA-IMVC under various missing rates (30%, 50% and 70%). The experimental results clearly demonstrate that the objective function value does not fluctuate drastically as the missing rate increases. This significantly supports our argument regarding the robustness of HA-IMVC in high-missing scenarios and proves that our proposed consensus anchor-guided hypergraph framework can maintain a stable optimization process by capturing high-order correlations, even when handling extremely sparse data.
>
> **Table 1: Objective values of HA-IMVC under different missing rates**
> | missing rate | 0.3 | 0.5 | 0.7 |
> | :--- | :--- | :--- | :--- |
> | **CCV** | 925.837 | 878.431 | 881.646 |
> | **BDGP** | 46.893 | 43.095 | 41.269 |
> | **NGs** | 2.992 | 2.831 | 3.274 |
> | **Caltech101-20** | 25.464 | 26.371 | 29.816 |
>
> Q3: Can you provide a motivating example where bipartite graph fail to capture high-order correlation between sample-anchor relations.
>
> A3: See A1.
>
> Q4: The F-score and silhouette score are also a classical metrics to measure clustering performance. Can you motivate the choice of focusing on the three chosen evaluation metrics (ACC, NMI, PUR)?
>
> A4: In this paper, we focus on Accuracy (ACC), Normalized Mutual Information (NMI), and Purity (PUR) as they are the most authoritative and universal performance evaluation standards in the fields of multi-view clustering and incomplete multi-view clustering. While F-score is also an important metric, its statistical properties in relevant literature are often highly correlated with ACC, making it less frequently reported than the other three metrics. Furthermore, as the Silhouette Score is an unsupervised internal evaluation metric that measures spatial separation rather than the accuracy of cluster assignments, we prioritize the three most representative external metrics to maintain the conciseness and focus of our experimental comparisons. If the paper is accepted, we will include F-score results in the revised manuscript to further validate the superiority of our algorithm.

---

> > ### Author Rebuttal · Reviewer_orK5 · 2026-04-01
> >
> > The rebuttal addressed all my concerns.

---

> > > ### Author Response · Authors · 2026-04-02
> > >
> > > Thank you very much for confirming that all concerns have been addressed. We are truly grateful for your time and the constructive suggestions, which have significantly improved our work.
> > >
> > > Since you have acknowledged that all issues are resolved, we would be very grateful if you could consider raising the score.
> > >
> > > Thank you again for your support.

---

### Decision · Program_Chairs · 2026-04-30

**Decision:**

Accept (regular)

**Comment:**

Although reviewers raised some concerns, during the discussion, the authors addressed all concerns, and the technical contributions and evaluations of this paper have been improved. At last, all reviewers agree on accepting this paper. Hence, my recommendation is Accept.